# On the numerical stability of surface-atmosphere coupling in weather and climate models

Anton Beljaars[1], Emanuel Dutra[1,2], Gianpaolo Balsamo[1], and Florian Lemarié[3]

[1]European Centre for Medium-Range Weather Forecasts, Shinfield Park, Reading, RG2 9AX, UK
[2]Instituto Dom Luiz, Faculdade de Ciências, Universidade de Lisboa, 1749-016 Lisboa, Portugal
[3]Inria, Univ. Grenoble Alpes, CNRS, LJK, Grenoble F-38000, France

*Correspondence to:* Anton Beljaars (anton.beljaars@ecmwf.int)

**Abstract.** Coupling the atmosphere with the underlying surface presents numerical stability challenges in cost-effective model integrations used for operational weather prediction or climate simulations. These are due to the choice of large integration time-step compared to the physical time scale of the problem, aiming at reducing computational burden, and to an explicit flux coupling formulation, often preferred for its simplicity and modularity. Atmospheric models therefore use the surface-layer temperatures (representative of the uppermost soil, snow, ice, water, etc.,) at the previous integration time-step in all surface-atmosphere heat-flux calculations and prescribe fluxes to be used in the surface model integrations. Although both models may use implicit formulations for the time stepping, the explicit flux coupling can still lead to instabilities.

In this study, idealized simulations with a fully coupled implicit system are performed to derive an empirical relation between surface heat flux and surface temperature at the new time level. Such a relation mimics the fully implicit formulation by allowing to estimate the surface temperature at the new time level without solving the surface heat diffusion problem. It is based on similarity reasoning and applies to any medium with constant heat diffusion and heat capacity parameters. The advantage is that modularity of the code is maintained and that the heat flux can be computed in the atmospheric model in such a way that instabilities in the snow or ice code are avoided. Applicability to snow/ice/soil models with variable density is discussed, and the loss of accuracy turns out to be small. A formal stability analysis confirms that the parametrized implicit flux coupling is unconditionally stable.

## 1 Introduction

Coupling atmospheric models to the underlying surface model involves both scientific and technical issues. Models of the atmospheric circulation tend to be computer intensive and therefore often employ long time steps (up to one hour), which is a challenge for stability and accuracy (Beljaars et al., 2004; Lemarié et al., 2015). The turbulent diffusion part of these codes provides the coupling to the surface, has short physical time scales near the surface and therefore needs implicit numerics for stability. The surface may be vegetation, soil, snow, ice, or a combination of these in a tile scheme. Best et al. (2004) propose a coupling strategy to the surface that has a clean interface between atmosphere and surface code, and allows to include the surface or the top part of the surface in the implicit computations. This is often necessary for stability if the physical time scale of e.g. vegetation, soil, snow or ice surface is short compared to the model time step.

The ideal solution for stability is to combine the boundary layer heat diffusion and e.g. the snow or ice layer diffusion in a single implicit solver. This has been demonstrated in a series of papers describing developments in the ORCHIDEE model (Polcher et al., 1998; Ryder et al., 2016; Wang et al., 2013). The same method was used by Schulz et al. (2001) in the Hamburg model. However, not all models do have sufficient modularity of the code to make it practical. The complication of processes like phase changes and water percolation also require implementation of conserved variables to support full implicitness (Wang et al., 2013). The standard solution is to compute fluxes at the surface on the basis of the old time level surface temperature. It is often called "explicit flux coupling". To improve stability and accuracy West et al. (2016) recently proposed to move the flux coupling level one level down i.e. just below the surface. This has the advantage of including the fast responding surface layer in the fully implicit computations, which is beneficial for stability and accuracy.

Ongoing work at ECMWF on snow modelling raised similar issues. The existing single layer snow model (see e.g. Dutra et al., 2010), has already a minor stability issue when the snow layer becomes very thin, e.g. during the first snowfall in the season and at the final melt. This was addressed by introducing some empirical implicitness in the coupling by making an educated guess of the future snow temperature. Initial experimentation with a multilayer snow model (Dutra et al., 2012) showed even more frequent instabilities, so more implicitness in the coupling is required for stability.

In this paper, we propose a solution, that has the simplicity and modularity of the explicit flux coupling, but still has the stability of the fully implicit system. To derive simple solutions, the fully implicit coupled system is used as a reference. It is shown that the tri-diagonal set of equations corresponding to the discretized diffusion equation (for snow, ice or soil) can be converted to a relation between temperature and heat flux at the surface. The coefficients in this relation are then parametrized depending on properties of the medium, time step and vertical discretization. The coefficients are put in dimensionless form, which makes the empirical coefficients universal and applicable to any medium and any discretization.

The experimental environment in this paper is a simple model of a near surface air layer coupled to a snow pack by turbulent exchange. The atmosphere (e.g. at a height of $10\,m$, typical for atmospheric models) is assumed to have a diurnal cycle, and the response of temperature in the snow pack is considered. Although the following sections refer to snow only, the dimensionless framework ensures that the outcome is valid for any medium.

The following two sections (2 and 3) describe the equations for the discretized snow layer and the turbulent coupling between atmosphere and snow. Sections 4, 5 and 6 describe the numerical solution for an idealized diurnal cycle, the parametrization of the coefficients that relate heat flux and top layer snow temperature and the testing of the proposed scheme. Finally, the results and their applicability are briefly discussed in the concluding section. Also the implications of non-uniform snow density are discussed. The numerical solver and a formal stability analysis are described in Appendix A and B respectively.

## 2 Implicit numerical solution of the diffusion equation

We consider the diffusion equation for temperature in snow

$$\rho C \frac{\partial T}{\partial t} = \frac{\partial G}{\partial z} \,, \tag{1}$$

$$G = K \frac{\partial T}{\partial z} \,, \tag{2}$$

where $\rho \ (kg m^{-3})$ is density, $C \ (J kg^{-1} K^{-1})$ is heat capacity, $T \ (K)$ is temperature, $G \ (W m^{-2})$ is heat flux, and $K \ (W m^{-1} K^{-1})$ is the diffusion coefficient for heat. The boundary conditions are:

$$G = G_0 \ for \ z = 0 \,, \tag{3}$$

$$G = 0 \ for \ z \to -\infty \,. \tag{4}$$

For numerical stability with long time steps it is necessary to use an implicit scheme. With a vertical grid defined as in Fig. 1, the equation can be discretized as follows

$$(\rho C)j \frac{T_j^{n+1} - T_j^n}{\Delta t} = \frac{1}{\Delta z_j} \left( K_{j-1/2} \frac{T_{j-1}^{n+1} - T_j^{n+1}}{\Delta z_{j-1/2}} - K_{j+1/2} \frac{T_j^{n+1} - T_{j+1}^{n+1}}{\Delta z_{j+1/2}} \right) \,, \tag{5}$$

with boundary conditions

$$(\rho C)_1 \frac{T_1^{n+1} - T_1^n}{\Delta t} = \frac{1}{\Delta z_1} \left( G_0 - K_{1+1/2} \frac{T_1^{n+1} - T_2^{n+1}}{\Delta z_{1+1/2}} \right) \,, \tag{6}$$

$$(\rho C)_{NL} \frac{T_{NL}^{n+1} - T_{NL}^n}{\Delta t} = -\frac{1}{\Delta z_{NL}} \left( K_{NL-1/2} \frac{T_{NL-1}^{n+1} - T_{NL}^{n+1}}{\Delta z_{NL-1/2}} \right) \,. \tag{7}$$

This set of equations forms a tri-diagonal system, with diagonals A, B and C (the coefficients are defined in Appendix A). The matrix equations can be solved by successive elimination from the bottom upward such that the C-coefficients are replaced by zeros. At the same time, the equations are scaled to obtain B-coefficients that are equal to 1. Arriving at the top, it provides a solution for $T_1^{n+1}$. The solution for the other layers can be found by successive back-substitution of the temperatures going from top to bottom (see Appendix A for more details).

In case $G_0$ is not known, the elimination provides a linear relation between $G_0$ and $T_1^{n+1}$

$$T_1^{n+1} = \alpha G_0 + \beta \,. \tag{8}$$

This relation can be used to achieve fully implicit coupling with the air/surface interaction formulation.

## 3 Coupling to the lowest model level of the atmosphere

To focus on stability of the atmosphere surface coupling, it is assumed that the evolution of the near atmospheric temperature is known, e.g. as in standalone simulations of the land surface. However, this is not a limitation in full 3D models that typically use an implicit solver for the turbulent diffusion. In that case the atmospheric model will perform the downward elimination

process (the same way as described in Appendix B). The result is a linear relation between the $n+1$ temperature at the lowest atmospheric level and the surface heat flux, which can be used with the air/land interaction formulae described below to achieve fully implicit coupling.

With a prescribed air temperature, the heat flux into the snow layer can be related to the air / surface temperature difference in the following way

$$G_0 = \rho_a c_p C_H |U| (T_a - T_{sk}) , \tag{9}$$

where $G_0$ is the heat flux into the snow pack, $\rho_a$ is air density, $c_p$ is air heat capacity, $C_H$ is the transfer coefficient between the atmospheric level and the surface, $|U|$ is absolute wind speed, $T_a$ is air temperature, and $T_{sk}$ is temperature of the snow surface (skin temperature).

The coupling through a transfer coefficient is standard and represents the integral profile function according to Monin Obukhov (MO) similarity (see e.g. Brutsaert, 1982). The transfer coefficient in neutral conditions is related to the height of the atmospheric level, and the surface roughness lengths of momentum and heat

$$C_H = \frac{\kappa^2}{ln(z_a/z_{om})ln(z_a/z_{oh})} , \tag{10}$$

where $\kappa$ is the VonKarman constant (0.4), $z_a$ is the height of the atmospheric level, $z_{om}$ is the surface roughness length for momentum, and $z_{oh}$ is the surface roughness length for heat. Stability can be included by extending the logarithmic terms with the integral MO stability functions.

In the vertically discretized snow (see Fig. 1), the temperature of layer 1 is assumed to be at the midpoint which is different from the skin temperature. Therefore, the total conductivity between the atmosphere and the first snow layer ($\lambda_t$) is composed of two components: the turbulent transfer in the air above the surface ($\lambda_a$) and the conductivity of half of the top snow layer ($\lambda_{sk}$). The two conductivities are in parallel, because the inverse of conductivities (resistances) are in series, leading to the following formulation for the heat flux into the snow

$$G_0 = \lambda_t (T_a - T_1) , \tag{11}$$

with

$$\lambda_t = \frac{\lambda_a \lambda_{sk}}{\lambda_a + \lambda_{sk}} ,$$
$$\lambda_a = \rho_a c_p C_H |U| ,$$
$$\lambda_{sk} = \frac{2 K_{1-1/2}}{\Delta z_1} .$$

Two different time stepping procedures are considered:

    i. **Explicit flux coupling**. This is the traditional approach where the expression for the surface flux uses the previous time level of the surface temperature leading to the following discretization of equation (11)

$$G_0 = \lambda_t (T_a^{n+1} - T_1^n) . \tag{12}$$

With the explicit specification of the flux at the surface flux, the tridiagonal system can be solved directly.

ii. **Implicit flux coupling.** The discretization of equation (11) reads

$$G_0 = \lambda_t(T_a^{n+1} - T_1^{n+1}),\tag{13}$$

With this fully implicit formulation, the surface heat flux can not be specified explicitly, so it has to be found as part of the coupled atmosphere/surface system. For that purpose the tri-diagonal problem is solved in two steps. First, the elimination part is performed resulting in a solution for $\alpha$ and $\beta$ in equation (8). Together with equation (13), $T_1^{n+1}$ and $G_0$ can be computed:

$$T_1^{n+1} = \frac{\alpha \lambda_t T_a^{n+1} + \beta}{1 + \alpha \lambda_t},\tag{14}$$

$$G_0 = \frac{\lambda_t(T_a^{n+1} - \beta)}{1 + \alpha \lambda_t}.\tag{15}$$

Finally the entire temperature profile can be resolved by performing the back-substitution in the tri-diagonal solver.

## 4 Solutions with a simple multilayer snow model

In this section, solutions are considered for a $1\,m$ thick snow layer with constant heat capacity and heat diffusion coefficients. Idealized temperature forcing from the atmosphere is prescribed as a sinusoidal diurnal cycle. The choice of constants is documented in Table 1. The initial temperature profile at $t = 0$ is set to $-5^oC$, and a single sinusoidal diurnal cycle with an amplitude of $1^oC$ is imposed at the $10\,m$ level in the atmosphere

$$T_{10} = -5 + sin(\frac{2\pi t}{3600. * 24.}).\tag{16}$$

The simulations are performed with different uniform vertical discretizations and different time steps. Fig. 2 shows time series of the snow skin temperature (left column) and the ground heat flux (right column), with the two schemes. The fairly long time step of 3600 seconds is selected to illustrate stability and time truncation issues, and a short time step of 100 seconds for comparison. In the latter case time truncation errors are small for both schemes (convergence was verified). The three rows in Fig. 2 are for different vertical discretizations: 0.2, 0.02 and 0.002 $m$.

The first thing to note is that amplitude and phase of the skin temperature diurnal cycle only have a small dependence on vertical resolution. This is surprising because the amplitude of diurnal cycle of layer 1 with $\Delta z = 0.2\,m$ is only 20% of the amplitude with $\Delta z = 0.02\,m$. The reason that the skin temperature is still reasonable is due to the conductivity between the middle of the layer and the top (much lower with $\Delta z = 0.2\,m$ than with $\Delta z = 0.02\,m$). So at low vertical resolution, a substantial part of the temperature signal at the snow skin is due to the "interpolation" between air and middle of the first snow layer making use of the air conductivity ($\lambda_a$) and the snow conductivity of half the top layer ($\lambda_{sk}$). One might interpret this result as a justification for rather low vertical resolution. However, it should be realized that the forcing has the diurnal time scale only. With faster time scales e.g. due to moving clouds and frontal passages, a relatively thick near surface layer will not be able to respond.

Although it is impossible to draw general conclusions about accuracy from limited experimentation, we note that the fully implicit solution with $\Delta t = 3600\,s$ is very close to the short time step solution with $\Delta t = 100\,s$, so the long time step does not compromise accuracy in this case, although the time stepping is first order accurate only. However, the solution with explicit coupling deviates visibly from the implicit and very short time step solutions (compare the red solid curve in middle/left panel of Fig. 2 with the blue curve). Apparently, it is the mismatch of time levels in the flux computation that is detrimental to accuracy. The error is particularly visible as a phase error.

Finally, the explicit coupling turns out to be unstable for very thin snow layers (see lower panels in Fig. 2 for $\Delta z = 0.002$. Also for this case the long time step solution with implicit coupling is fairly accurate as it is very close to the short time step solution. These experimental results are confirmed by a formal stability analysis in Appendix B. The explicit flux coupling is unstable for a particular parameter range and the implicit flux coupling is unconditionally stable.

Because of the good stability and accuracy characteristics, we develop in the next section a parametric form of $\alpha$ and $\beta$ in equation (8).

## 5 Scaling relations for $\alpha$ and $\beta$

As suggested above, it is desirable to have all the flux formulations (also for the atmosphere/surface exchange) at the new time level $n + 1$. This implies the fully implicit option as suggested by (Polcher et al., 1998) and described in sections 2 and 3. It also requires to perform the elimination part of the tri-diagonal solver to find the relation between $T_1^{n+1}$ and $G_0$ according to equation (8). For code technical reasons it is often desirable to compute the heat flux into the snow, before the snow code is actually executed. Therefore, an educated guess is made of the coefficients $\alpha$ and $\beta$ in equation (8) without solving the tri-diagonal system, i.e. $\alpha$ and $\beta$ are parametrized.

For that purpose, we make use of similarity theory for the diffusion equation with constant coefficients. If we think of an infinite medium (thick snow layer) with uniform temperature $T_o$ and make a jump at the surface to $T_{new}$ at $t = 0$, we have to consider the following basic variables: the temperature change $T - T_0$ at time $t$, $T_{new} - T_0$, $K/(\rho C)$, and depth $z$. According to the Buckingham Pi Theorem (Stull, 1988), 5 variables with 3 dimensions ($m$, $s$, and $K$), lead to two independent dimensionless groups: $(T - T_0)/(T_{new} - T_0)$ and $z/\delta$, where

$$\delta = \left( \frac{Kt}{\rho C} \right)^{1/2}. \tag{17}$$

Length scale $\delta$ is the natural length scale of the medium for time scale $t$ after which the temperature change at the surface was applied. From the physical point of view, $\delta$ is the typical depth to which the perturbation of the surface temperature has propagated at time $t$. The implication is that $(T - T_0)/(T_{new} - T_0)$ is a universal function of $z/\delta$. At this stage we do not care about the form, although the solution can be found easily by transforming the equation to the new coordinate $z/\delta$, which allows to separate the time dependence and the depth dependence leading to an ordinary differential equations which can be solved analytically (Carslaw and Jaeger, 1959).

Similarly, we can apply an external forcing by suddenly applying a heat flux $G_0$ at time 0 and look for the temperature response. Instead of scaling the temperature with the temperature jump, we make the temperature change dimensionless with $G_0$ and obtain

$$\frac{K(T-T_0)}{\delta G_0} = h\left(\frac{z}{\delta}\right) \text{ , or } T = \frac{\delta G_0}{K}h\left(\frac{z}{\delta}\right) + T_0 \text{ ,} \tag{18}$$

where $h$ is a universal function. For $z = 0$, equation (18) is of the form of equation (8). With time scale $\Delta t$ and substitution of the expression for $\delta$, we therefore expect the following scaling behavior for $\alpha$

$$\alpha \sim \left(\frac{\Delta t}{K \rho C}\right)^{1/2} \text{ .} \tag{19}$$

It indicates the surface temperature response to a $1\ W/m^2$ heat flux forcing over a finite time step $\Delta t$.

The scaling arguments above apply to the continuous system. For the discretized system, the scaling behavior of $\alpha$ also
depends on $\Delta z$, which introduces a dependence on the dimensionless variable $\Delta z/\delta$. For a very fine grid ($\Delta z << \delta$), the discrete system behaves like the continuous system and equation (19) applies. For a very thick top layer ($\Delta z >> \delta$), the heat flux is simply distributed over the top layer and the following applies

$$\alpha = \frac{\Delta t}{\Delta z \rho C} \text{ .} \tag{20}$$

In general the dimensionless $\alpha$ should be a universal function of $\delta/\Delta z$, i.e.

$$\alpha \left(\frac{K \rho C}{\Delta t}\right)^{1/2} = f\left(\frac{\delta}{\Delta z}\right) = f\left(\frac{(K\Delta t)^{1/2}}{\Delta z\,(\rho C)^{1/2}}\right) \text{ .} \tag{21}$$

The empirical function can be "measured" by running the numerical model as in the previous section for a range of time steps and vertical discretizations. Note that $\alpha$ remains constant during the time stepping and does not depend on the temperature profile. It is just a property of the tri-diagonal matrix which only contains properties of the medium, the time step and the level thickness. The results are shown in Fig. 3. Time steps range from $100\ s$ to $3600\ s$, and layer thicknesses are used from $0.002$
$m$ to $0.2\ m$, with a total snow depth of $1\ m$ for all simulations

For small ratios of $\delta/\Delta z$, the universal function should scale with equation (20) and for large values with (19). Surprisingly, coefficient $h_0$ turns out to be 1. An empirical fit is proposed that makes a smooth transition between the two regimes according to (see Fig. 3)

$$f(x) = \frac{x}{(1+x^{1.3})^{1/1.3}} \text{ .} \tag{22}$$

The exponent of 1.3 has been optimized to obtain a reasonable representation of the numerical data in the transition regime.

The second parameter for which an empirical formulation is needed is $\beta$. The physical meaning of $\beta$ is clear from equation (8): it is the temperature of the top snow layer at the new time level $T_1^{n+1}$ in case of zero heat flux. A simple approximation would be to select the temperature of the previous time level, but this is only valid for a uniform temperature profile. For a non-uniform temperature profile, heat diffusion will homogenize temperature, which will make $\beta$ different from $T_1^n$ at the old
time level. Following the scaling arguments above, we know that information propagates vertically over a distance $\delta$ during

time step $\Delta$. Therefore, we conjecture that the temperature of the old profile at depth $\delta$ is a better approximation for $\beta$ than the temperature at level 1, i.e. $T_\delta^n$ is better than $T_1^n$. Fig. 4 indeed confirms that the temperature at depth $\delta$ is a reasonable approximation. The temperature at $z = -\delta$ has been obtained by linear interpolation between levels, except when $\delta < 0.5\Delta z$. In the latter case, temperature $T_1^n$ is selected. Note that, unlike $\alpha$, $\beta$ does change with temperature and does evolve during the integration.

From Figs. 3 and 4, it is concluded that reasonable estimates can be made of $\alpha$ and $\beta$ without actually solving the tri-diagonal matrix. Depth scale $\delta$ and the thickness of the top layer $\Delta z$ are crucial scales to characterize the temperature evolution of the top snow layer over a time step.

## 6  Simulations with the empirical formulation

With the empirical formulations for $\alpha$ and $\beta$, it is possible now to repeat the simulations of section 4. Instead of generating the fully implicit solution by solving the tri-diagonal matrix in the standard way, $\alpha$ and $\beta$ are replaced by the empirical formulation between the elimination and back-substitution phase. If the formulation is perfect, the solution should be the same as the fully implicit solution. Results are shown in Fig. 5 for the skin temperature and the heat flux. Layer thicknesses of 0.2, 0.02 and 0.002 $m$ are shown as different rows in Fig. 5. The figure confirms that the diurnal temperature cycle of the fully implicit solution (blue curve, IMPL) is well reproduced by the solution with parametrized $\alpha$ and $\beta$ (black solid cure, IMPPAR). The differences between blue and black curves are very small.

Finally, the scheme was further simplified by using the parametric form for $\alpha$ only and estimating $\beta$ by putting it equal to $T_1^n$. The advantage is that no interpolation to $z = -\delta$ is needed, but that stability of the coupling is still maintained. However, it is clear that numerical errors are increased for thin snow layers (see dashed black curve). Such errors have to be seen in the context of other model errors, so the use of a parametrized $\alpha$ only, to ensure stability, may still be sufficient for many applications.

## 7  Discussion and conclusion

Numerical stability is a critical issue for atmospheric models that are coupled to a fast responding surface e.g. through a thin snow or ice layer. Very thin snow layers can occur in early winter after the first snow fall and during melt in spring. A fine discretization may also be desirable to allow for a fast response of the surface temperature to changes in radiation. Formal stability analysis confirms that unconditional stability can be achieved by a fully implicit coupling between atmosphere and surface.

Fully implicit coupling leads to a tri-diagonal problem in which atmosphere and surface are solved simultaneously. In practice, often so-called explicit flux coupling is applied: the atmospheric model uses the surface temperature of the previous time level to compute the surface heat flux, which is used later as boundary condition for the heat diffusion in the surface.

Explicit surface coupling puts stability limits on the thickness of the top snow layer and on the time step. Explicit flux coupling is often applied, because existing codes do not necessarily have sufficient modularity to support fully implicit coupling.

Although the atmosphere / surface heat diffusion leads to a single tri-diagonal matrix problem, one can also break it up in different steps. It is shown that the elimination part of the solver of the snow heat diffusion problem leads to a linear relation between surface temperature and surface heat flux. This relation can be used together with the atmosphere / surface interaction formulation to solve for the surface heat flux.

A simple method has been developed to approximate the coefficients in this linear relation. The coefficients are scaled with the characteristic scales of the diffusion equation. This makes the result universal and applicable to an arbitrary medium e.g. snow, ice or soil. The depth scale that characterizes the penetration of a perturbation over a time step, turns out to play a crucial role. In this paper the relevant empirical function is "measured" by solving the diffusion equation for a range of vertical resolutions and time steps.

Finally, the empirical functions are used to solve for the coupled diffusion problem and compared with the fully implicit computations. The results are very close. The advantage of the method is that the surface fluxes can be computed without calling any surface code, and behaves like explicit flux coupling. The only difference is that the surface heat flux expression has a damping term depending on the time step. This damping term is the result of the change of surface temperature related to the heat flux, and stabilizes the result.

The scaling argument used above only applies for a diffusion equation with constant properties of the medium. However, in reality there may be a profile of e.g. snow density as snow becomes more and more compact in deeper layers, or vertical resolution may be variable. The latter is numerically equivalent to a variable diffusion coefficient [1]. As a simple test, a case was selected where the profile of density is 150 $kgm^{-3}$ at the surface, increases linearly to 250 $kgm^{-3}$ at a depth of 0.5 $m$, and remains constant below 0.5 $m$. The characteristic depth is again computed as in section 5, and to non-dimensionalize, the snow properties are taken from the middle of the top snow layer. For this case the dimensionless $\alpha$ and characteristic temperature $\beta$ are shown in Figs. 6 and 7. They are very close to the figures for constant snow properties (Figs. 3 and 4), which suggests that the sensitivity to snow properties is fairly small. In general, it is to be expected that the snow properties very close to the surface control the relation between flux and temperature over a short time step, because the penetration depth $\delta$ is small.

We conclude that making an estimate of the relation between heat flux and surface temperature is a practical solution to support explicit flux coupling and to combine numerical stability for long time steps with a modular code structure. A formal stability analysis in Appendix B confirms unconditional stability of the proposed coupling method. The similarity framework makes the method applicable to any medium, e.g. snow, ice or soil. It is also worth noting that the method does not compromise conservation: the heat flux that is computed by the atmospheric model is later used by the surface model as boundary condition.

---

[1]In fact the aerodynamic coupling between atmosphere and snow can be interpreted as a big jump in the properties of the medium

## Appendix A: Solving the tri-diagonal matrix equations

The set of equations discussed in section 2 leads to the following tri-diagonal system

$$
\begin{pmatrix}
B_1 & C_1 & 0 & 0 & \cdots & 0 \\
A_2 & B_2 & C_2 & 0 & \cdots & 0 \\
0 & A_3 & B_3 & C_3 & \cdots & 0 \\
\ddots & \ddots & \ddots & \ddots & \ddots & \ddots \\
0 & \cdots & A_{NL-2} & B_{NL-2} & C_{NL-2} & 0 \\
0 & \cdots & 0 & A_{NL-1} & B_{NL-1} & C_{NL-1} \\
0 & \cdots & 0 & 0 & A_{NL} & B_{NL}
\end{pmatrix}
\begin{pmatrix}
T_1^{n+1} \\
T_2^{n+1} \\
T_3^{n+1} \\
\vdots \\
T_{NL-2}^{n+1} \\
T_{NL-1}^{n+1} \\
T_{NL}^{n+1}
\end{pmatrix}
=
\begin{pmatrix}
R_1 \\
R_2 \\
R_3 \\
\vdots \\
R_{NL-2} \\
R_{NL-1} \\
R_{NL}
\end{pmatrix}
\tag{A1}
$$

where

$$
\begin{aligned}
A_j &= -\frac{K_{j-1/2}}{\Delta z_j \, \Delta z_{j-1/2}} \,, \\
B_j &= \frac{(\rho C)_j}{\Delta t} + \frac{K_{j-1/2}}{\Delta z_j \, \Delta z_{j-1/2}} + \frac{K_{j+1/2}}{\Delta z_j \, \Delta z_{j+1/2}} \,, \\
C_j &= -\frac{K_{j+1/2}}{\Delta z_j \, \Delta z_{j+1/2}} \,, \\
R_j &= \frac{(\rho C)j}{\Delta t} T_j^n \,,
\end{aligned}
\tag{A2}
$$

with boundary condition at the surface

$$
\begin{aligned}
A_1 &= 0 \,, \\
B_1 &= \frac{(\rho C)_1}{\Delta t} + \frac{K_{1+1/2}}{\Delta z_1 \, \Delta z_{1+1/2}} \,, \\
C_1 &= -\frac{K_{1+1/2}}{\Delta z_1 \, \Delta z_{1+1/2}} \,, \\
R_1 &= \frac{G_0}{\Delta z_1} + \frac{(\rho C)_1}{\Delta t} T_1^n \,,
\end{aligned}
\tag{A3}
$$

and the no-flux condition at the bottom

$$
\begin{aligned}
A_{NL} &= -\frac{K_{NL-1/2}}{\Delta z_{NL} \, \Delta z_{NL-1/2}} \,, \\
B_{NL} &= \frac{(\rho C)_{NL}}{\Delta t} + \frac{K_{NL-1/2}}{\Delta z_{NL} \, \Delta z_{NL-1/2}} \,, \\
C_{NL} &= 0 \,, \\
R_{NL} &= \frac{(\rho C)_{NL}}{\Delta t} T_{NL}^n \,.
\end{aligned}
\tag{A4}
$$

The tridiagonal system is solved in two steps by standard Gaussian elimination. The first step is an upward sweep to eliminate the $C$ coefficients. It starts at level $NL$ by rescaling coefficient $A_{NL}$ to 1. The new coefficients identified by superscript * are

$$
A_{NL}^* = 1 \,, \quad B_{NL}^* = \frac{B_{NL}}{A_{NL}} \,, \quad R_{NL}^* = \frac{R_{NL}}{A_{NL}} \,.
\tag{A5}
$$

Next, coefficient $C_{NL-1}$ is eliminated by multiplying $A, B, C$ and $R$ for level $NL-1$ by $B_{NL}^*$, the new coefficients for level $NL$ with $C_{NL-1}$ (i.e. from A5), subtracting the two equations and rescaling the result to have 1 at the position of $A_{NL-1}$. The new $A$, $B$, $C$ and $R$ for level $NL-1$ are

$$A_{NL-1}^* = 1\,, \ \ B_{NL-1}^* = \frac{B_{NL-1}B_{NL}^* - C_{NL-1}}{A_{NL-1}B_{NL}^*}\,, \ \ C_{NL-1}^* = 0\,, \ \ R_{NL-1}^* = \frac{R_{NL-1}B_{NL}^* - C_{NL-1}R_{NL}^*}{A_{NL-1}B_{NL}^*}\,. \tag{A6}$$

This process is repeated to the top, which results in a matrix where the $A$-diagonals are all 1 and the $C$-diagonal contains zeros. If the surface heat flux is specified, the top line of the matrix contains the solution for $T_1^{n+1}$. The temperatures of all the other levels can be computed in a downward sweep, where the temperature of level $j$ is used to find the solution for level $j+1$ with the equation for level $j+1$.

If the surface heat flux is not known, the first line of the matrix equation contains a linear relation between $T_1^{n-1}$ and $G_0$,
which can be written in the form of equation (8).

## Appendix B: Stability analysis of the coupling schemes

In this section we present the stability properties of the three coupling methods introduced in the present study, namely the explicit flux coupling (EXPFLX), the implicit flux coupling (IMPFLX) and the so-called parametrized implicit flux coupling (IMPPAR). Since the numerical stability is expected to be greatly influenced by the numerical treatment of the surface boundary
condition, a classical Von Neumann stability analysis, which assumes periodic boundary conditions, would not be adequate. For this reason our study is based on a matrix stability analysis (e.g. Oishi et al., 2008). Assuming a constant grid spacing and diffusion coefficients, the results are shown in terms of the dimensionless coefficients $\sigma$ and $\gamma$ defined as

$$\sigma = \left(\frac{\delta}{\Delta z}\right)^2 = \frac{K\Delta t}{(\rho C)\Delta z^2} \qquad \gamma = \frac{\lambda_t \Delta t}{(\rho C)\Delta z}, \tag{B1}$$

where $\lambda_t$ and $\delta$ are respectively defined in (11) and (17), with $t = \Delta t$. The typical value of those parameters for the numerical
simulations discussed in sections 4 and 6 are given in table 2.

### B1   Numerical treatment of the surface boundary condition

Without loss of generality we consider in the following that $T_a = 0$ in the computation of the surface boundary condition $G_0$ defined in (11) as well as that $K$, $\Delta z$, and $\rho C$ are held constant. The only difference between the three coupling algorithms considered here is in the treatment of the surface boundary condition (i.e. for the vertical index $j = 1$):

– *Explicit flux coupling:* the surface temperature involved in the computation of $G_0$ is $T_1^n$, thus leading to the following counterpart of equation (6)

$$(1+\sigma)\,T_1^{n+1} - \sigma T_2^{n+1} = (1-\gamma)T_1^n \tag{B2}$$

– *Implicit flux coupling:* the surface temperature at time level $n+1$ is used to compute $G_0$

$$(1+\sigma+\gamma)\,T_1^{n+1} - \sigma T_2^{n+1} = T_1^n\,. \tag{B3}$$

– *Parametrized implicit flux coupling*: the temperature at time level $n+1$ is diagnosed as $T_1^n/(1+\alpha\lambda_t)$

$$(1+\sigma)\,T_1^{n+1} - \sigma T_2^{n+1} = \left(1 - \frac{\gamma}{1+\alpha\lambda_t}\right) T_1^n \, . \tag{B4}$$

where $\alpha$ is defined in (21). Using (22) it can readily be seen that

$$\frac{\gamma}{1+\alpha\lambda_t} = \frac{\gamma}{1+\gamma(1+\sqrt{\sigma}^{1.3})^{-1/1.3}} \, , \tag{B5}$$

which shows that the parametrized implicit flux coupling can be interpreted as a limiter acting on the value of $\gamma$ of the explicit flux coupling, indeed $\gamma(1+\alpha\lambda_t)^{-1} \le \gamma$.

## B2   Matrix stability analysis

As shown in Appendix A, the Euler implicit scheme applied to the diffusion equation can be written in a general matrix form

$$\mathbf{A}\mathbf{T}^{n+1} = \mathbf{B}\mathbf{T}^n, \qquad \mathbf{T} = (T_1, \dots, T_{NL})^t, \tag{B6}$$

with

$$\mathbf{A} = \begin{pmatrix} 1+\sigma+\theta(2\theta-1)\gamma & -\sigma & 0 & 0 & \dots \\ -\sigma & 1+2\sigma & -\sigma & 0 & \dots \\ 0 & \ddots & \ddots & \ddots & 0 \\ \dots & 0 & -\sigma & 1+2\sigma & -\sigma \\ \dots & 0 & 0 & -\sigma & 1+\sigma \end{pmatrix}, \tag{B7}$$

$$\mathbf{B} = \begin{pmatrix} 1+(\theta-1)(1+2\frac{1-\alpha\lambda_t}{1+\alpha\lambda_t}\theta)\gamma & 0 & 0 & \dots & 0 \\ 0 & 1 & 0 & \dots & 0 \\ 0 & & \ddots & \ddots & \ddots & 0 \\ \dots & & 0 & 0 & 1 & 0 \\ \dots & & 0 & 0 & 0 & 1 \end{pmatrix}, \tag{B8}$$

where $\theta = 0$ corresponds to the explicit flux coupling, $\theta = 1$ to the implicit flux coupling, and $\theta = 1/2$ to the parameterized implicit flux coupling. The general implicit scheme (B6) is stable if all the eigenvalues of the matrix $\mathbf{M} = \mathbf{A}^{-1}\mathbf{B}$ do not exceed 1 in magnitude. Therefore, the stability analysis requires the computation of the spectral radius of matrix $\mathbf{M}$, i.e. its larger eigenvalue in magnitude. For $\gamma \ge 0$ it can be shown that the smallest eigenvalue of $\mathbf{A}$ is larger or equal[2] to 1 meaning that this matrix is invertible for $\theta \in \{0, 1/2, 1\}$. In Fig. 8, values of the spectral radius of $\mathbf{M}$ obtained over a range of values of $\gamma$

---

[2]for the special cases $\theta = 0$ and $\theta = 1/2$, the eigenvalues $\lambda_k^{\mathbf{A}}$ of matrix $\mathbf{A}$ are given by $\lambda_k^{\mathbf{A}} = 1 + 2\sigma\left(1 + \cos\frac{k\pi}{NL}\right)$ for $k = 1, \dots, NL$; therefore $\lambda_{\min}^{\mathbf{A}} = 1, \forall \sigma \ge 0$.

and $\sigma$ are shown for each coupling algorithm[3]. Gray shaded areas coincide with regions where the spectral radius is larger than 1 thus indicating parameter values for which the corresponding scheme is unstable. From those results, the only algorithm that turns out to be conditionally stable is the explicit flux coupling whereas the implicit and parameterized implicit flux coupling are unconditionally stable. The results are thus consistent with the numerical experiments discussed in sections 4 and 6.

Empirically, it can be found that the stability condition for the explicit flux coupling roughly behaves like $\gamma \leq 2 + \sqrt{\sigma}^{1.1}$ (see figure 9). For the parametrized implicit flux coupling, $\gamma$ is replaced by $\widetilde{\gamma}(\sigma) = \frac{\gamma}{1+\gamma(1+\sqrt{\sigma}^{1.3})^{-1/1.3}}$ which is always smaller than $\widetilde{\gamma}_{\max} = (1 + \sqrt{\sigma}^{1.3})^{1/1.3}$. As shown in figure 9, $\forall \sigma \geq 0$, $\widetilde{\gamma}_{\max} \leq 2 + \sqrt{\sigma}^{1.1}$ meaning that for the particular choice of $f(\sqrt{\sigma})$ given in (22) the parameterized implicit flux coupling is unconditionally stable because it always satisfies the stability constraint of the explicit flux coupling.

**Data availability**

The data that is used in this paper has been produced with a dedicated stand-alone fortran program. ECMWF's data policy does not allow open access to software. However, the code can be obtained from the first author subject to license. The license implies non-commercial use i.e. for research and education only.

**Acknowledgements**

The authors would like to thank Alex West and Jan Polcher for their comprehensive reviews and for suggesting numerous improvements to the manuscript. F. Lemarié acknowledges the support of the French National Research Agency (ANR) through contract ANR-16-CE01-0007. We also thank Erland Källén and Nils Wedi for careful reading of an early version of the manuscript.

---

[3]Numerical results are obtained for $NL = 50$ after checking that an increased number of vertical levels does not change the results significantly. The use of very few vertical levels ($N \leq 10$) could lead to different stability results because only very few eigenmodes will be properly resolved. However, in this case we do not expect major stability issues since the values of $\sigma$ and $\gamma$ are very small, see table 2.

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

**Table 1.** List of parameters used in the idealized simulation of a snow layer

| Parameter | Description | Value | Units |
|---|---|---|---|
| $\rho$ | snow density | 150 | $kg\,m^{-3}$ |
| $\rho_{ice}$ | ice density | 920 | $kg\,m^{-3}$ |
| $C$ | snow (and ice) heat capacity | 2228 | $J\,kg^{-1}K^{-1}$ |
| $K_{ice}$ | ice heat diffusion coefficient | 2.2 | $W\,m^{-1}K^{-1}$ |
| $K$ | snow heat diffusion coefficient | $K_{ice}(\rho/\rho_{ice})^{1.88}$ | $W\,m^{-1}K^{-1}$ |
| $\rho_a$ | air density | 1.2 | $kg\,m^{-3}$ |
| $c_p$ | air heat capacity | 1005 | $J\,kg^{-1}K^{-1}$ |
| $|U|$ | absolute wind speed | 4 | $m\,s^{-1}$ |
| $z_{om}$ | roughness length for momentum | 0.0001 | $m$ |
| $z_{oh}$ | roughness length for heat | 0.0001 | $m$ |
| $z_a$ | height atmospheric forcing level | 10 | $m$ |
| $\kappa$ | VonKarman constant | 0.4 | $-$ |
| $D$ | total depth of snow layer | 1 | $m$ |

**Table 2.** Values of $\gamma$ and $\sigma$ defined in (B1) for various time steps $\Delta t$ and vertical discretizations $\Delta z$, for $\rho = 150\,kg m^{-3}$, $C = 2228\,J\,kg^{-1}K^{-1}$, and $K = K_{ice}(\rho/\rho_{ice})^{1.88} \approx 7.27 \times 10^{-2}\,W m^{-1}K^{-1}$.

| $\Delta t$ | $[s]$ | 100 | 3600 | 100 | 3600 | 100 | 3600 |
|---|---|---|---|---|---|---|---|
| $\Delta z$ | $[m]$ | 0.2 | 0.2 | 0.02 | 0.02 | 0.002 | 0.002 |
| $\lambda_t$ | $[W m^{-2}K^{-1}]$ | 0.65 | 0.65 | 3.23 | 3.23 | 5.39 | 5.39 |
| $\gamma$ | $[-]$ | $9.6 \times 10^{-4}$ | $3.48 \times 10^{-2}$ | $4.81 \times 10^{-2}$ | 1.74 | 0.8 | 29 |
| $\sigma$ | $[-]$ | $5.44 \times 10^{-4}$ | $1.96 \times 10^{-2}$ | $5.44 \times 10^{-2}$ | 1.96 | 5.44 | 195.8 |

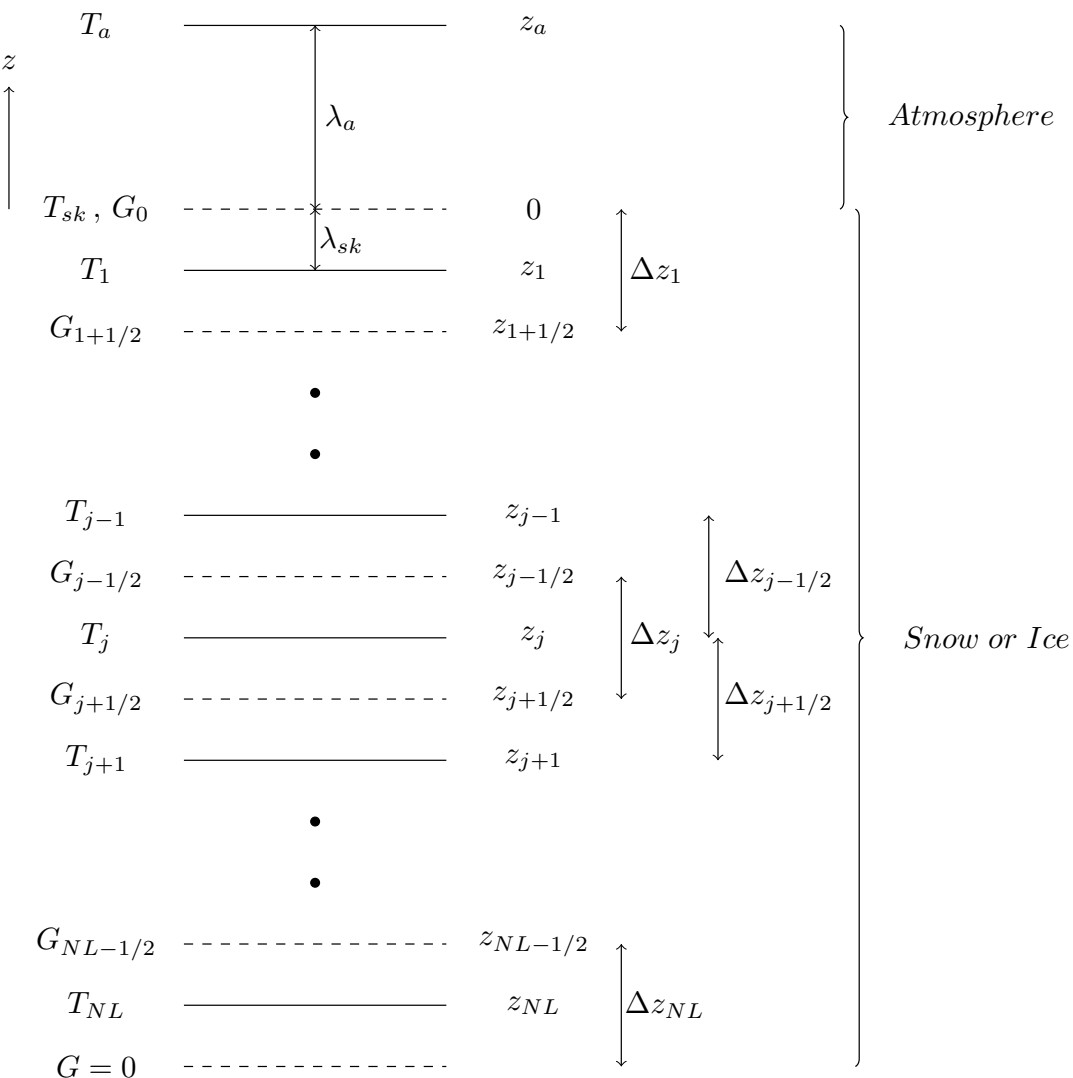

**Figure 1.** The numerical grid is defined by the position of the half levels, i.e. the thickness of the layers. The full levels are in the middle of the layers, i.e. $z_j = (z_{j-1/2} + z_{j+1/2})/2$. The surface is at $z = 0$. The bottom level is defined by the accumulated depth of all the layers. The temperature is defined on full levels and the heat fluxes are defined on half levels.

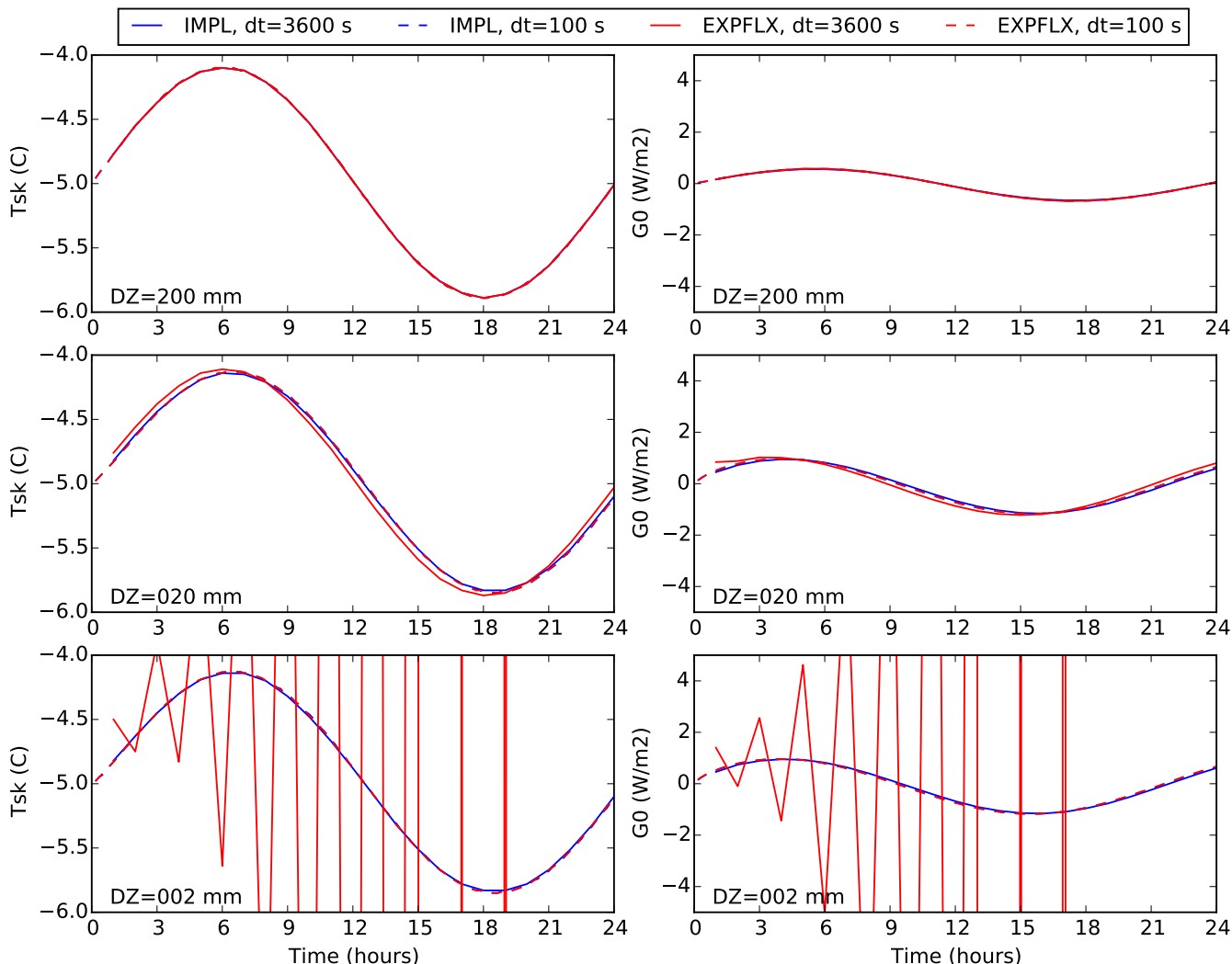

**Figure 2.** Diurnal cycle time series of snow skin temperature (left column) and surface heat flux (right column). The simulations were made with 0.2, 0.02 and 0.002 $m$ vertical resolution (top, middle and bottom panels). The blue curves refer to the fully implicit solution(IMPL); the red curves indicate the solutions with explicit flux coupling (EXPFLX). The solid curves are with a time step of 3600 $s$ seconds and the dashed curves with 100 seconds.

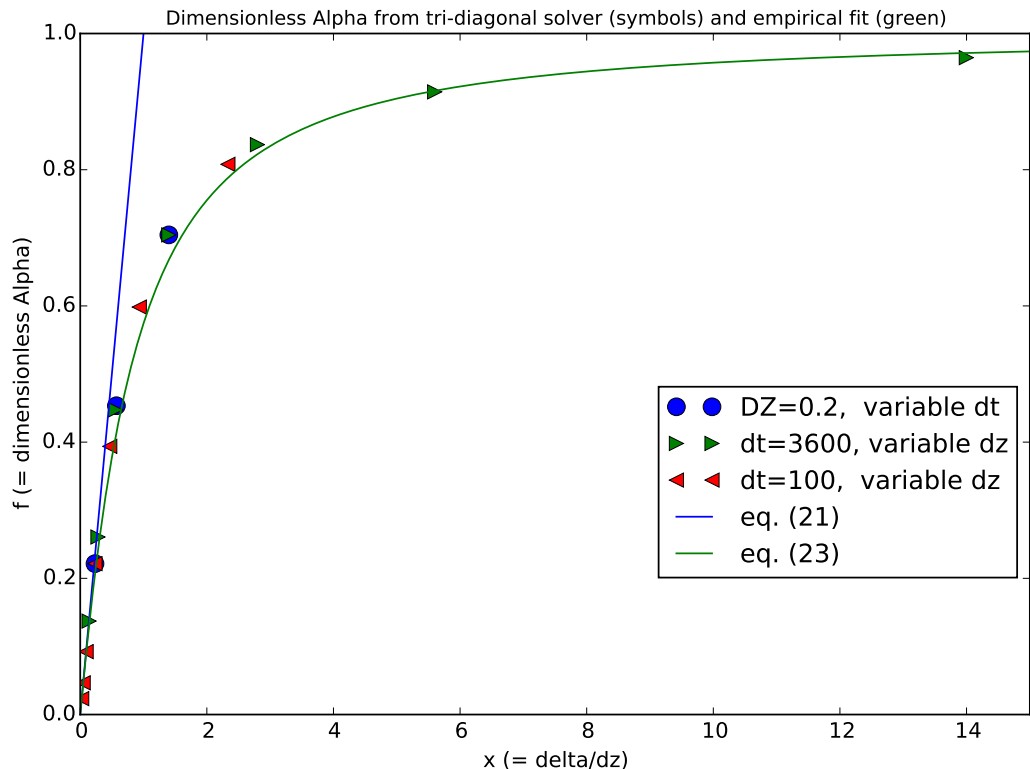

**Figure 3.** Dimensionless function $f = \alpha(K \, \rho C / \Delta t)^{1/2}$ as a function of $x = \delta/\Delta z$. The circles and triangles are for different combinations of $\Delta z$ and $\Delta t$. The blue line is the asymptotic limit for small $\delta/\Delta z$. The green curve is the empirical fit according to equation (22).

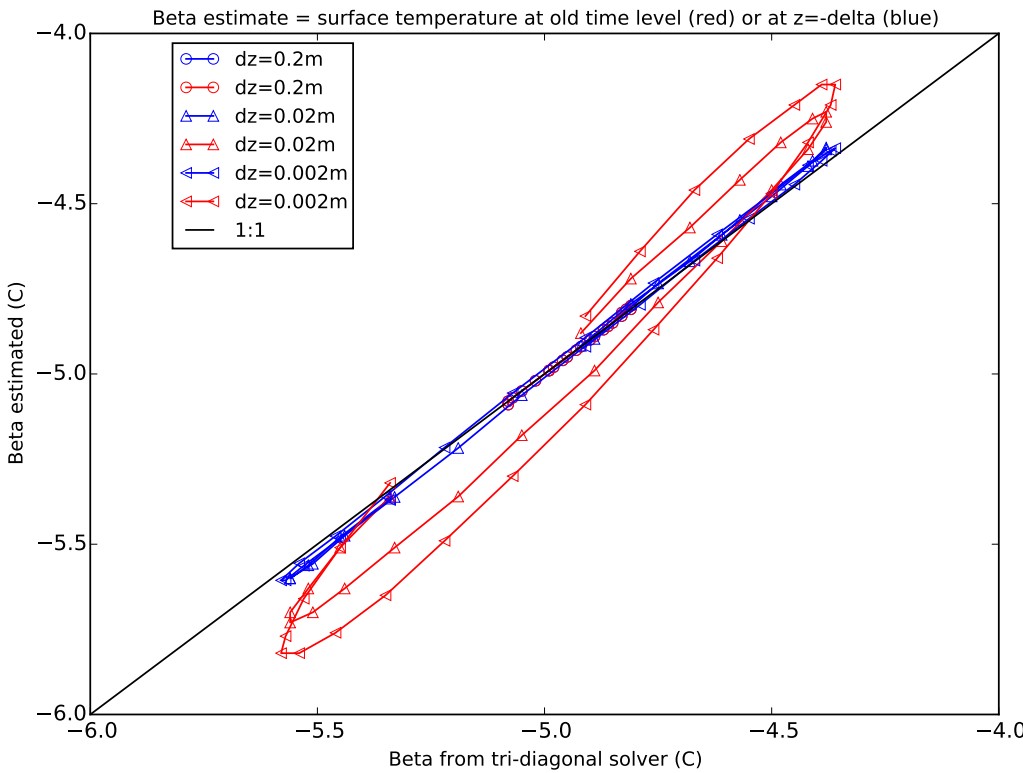

**Figure 4.** Empirical estimates of parameter $\beta$ as a function of the value found from the tri-diagonal solver. The red curve represents the estimate according to $T_1^n$ and the blue curve is the temperature at $z = -\delta$, also at the previous time level $n$. The symbols (connected by lines) indicate the successive time steps in the diurnal cycle. Results are plotted for vertical resolutions of 0.2, 0.02 and 0.002 $m$.

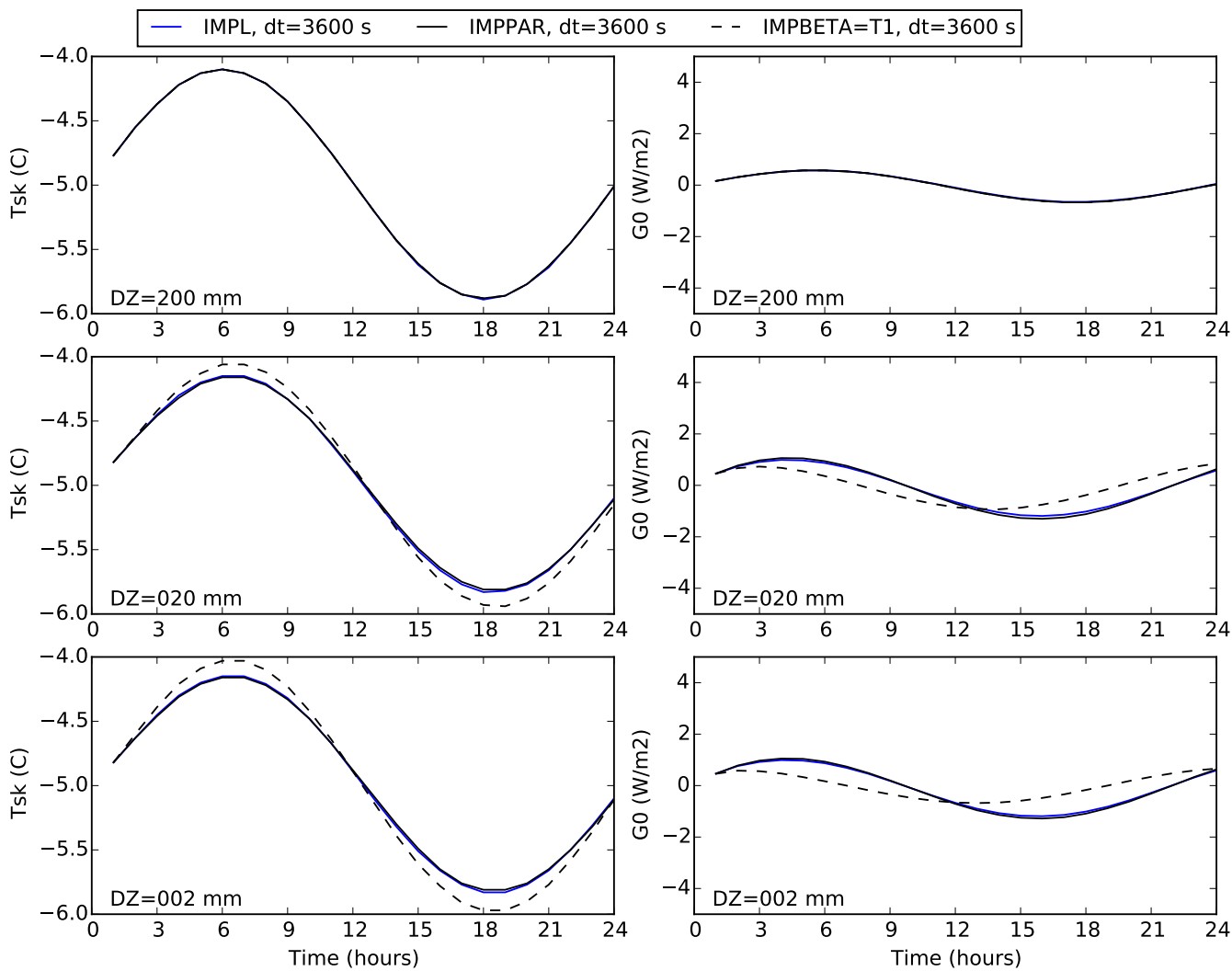

**Figure 5.** Diurnal cycle series of skin temperature (left columns and) and surface heat flux (right columns). The simulations were made with 0.2, 0.02 and 0.002 $m$ resolution (top, middle and bottom panels). The blue curve refers to the fully implicit solution(IMPL); the black solid curve is the solution with parametrized $\alpha$ and $\beta$. The black dashes curve refers to the solution where $\alpha$ is parametrized and $\beta$ is set equal to the temperature of level 1 at the previous time (n). The time step is 3600 seconds.

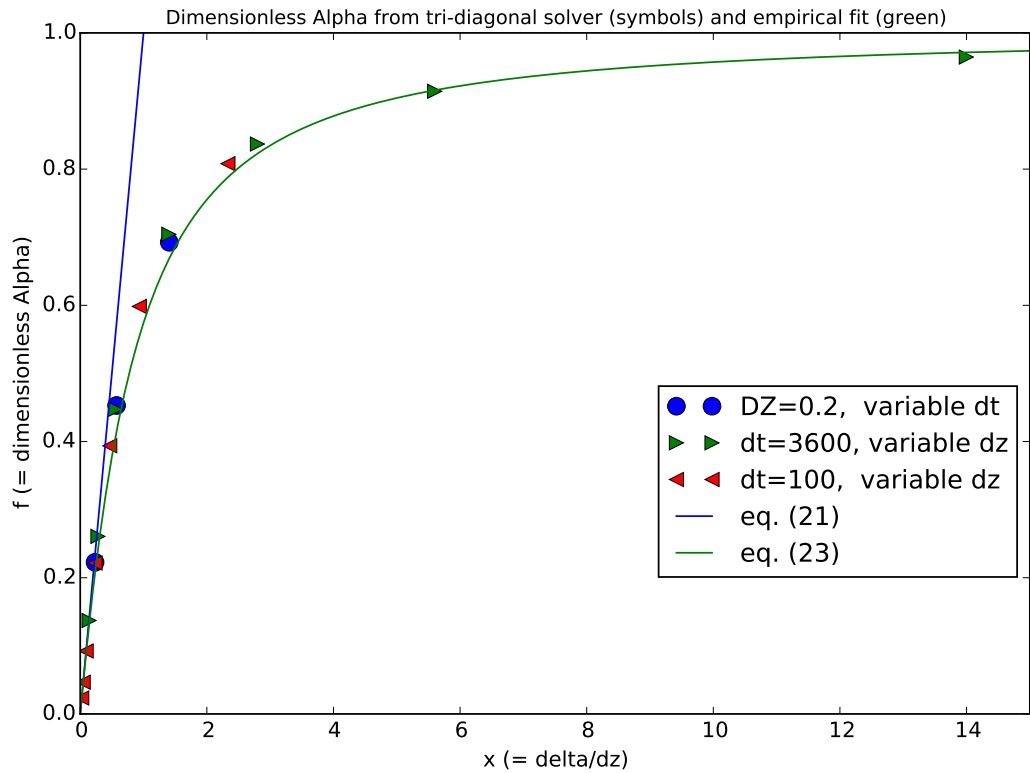

**Figure 6.** Dimensionless $\alpha$ as in Fig. 3, but for non-uniform snow density. The snow density is $150\ kgm^{-3}$ at the surface, increases linearly to $250\ kgm^{-3}$ at a depth of $0.5\ m$, and remains constant below $0.5\ m$.

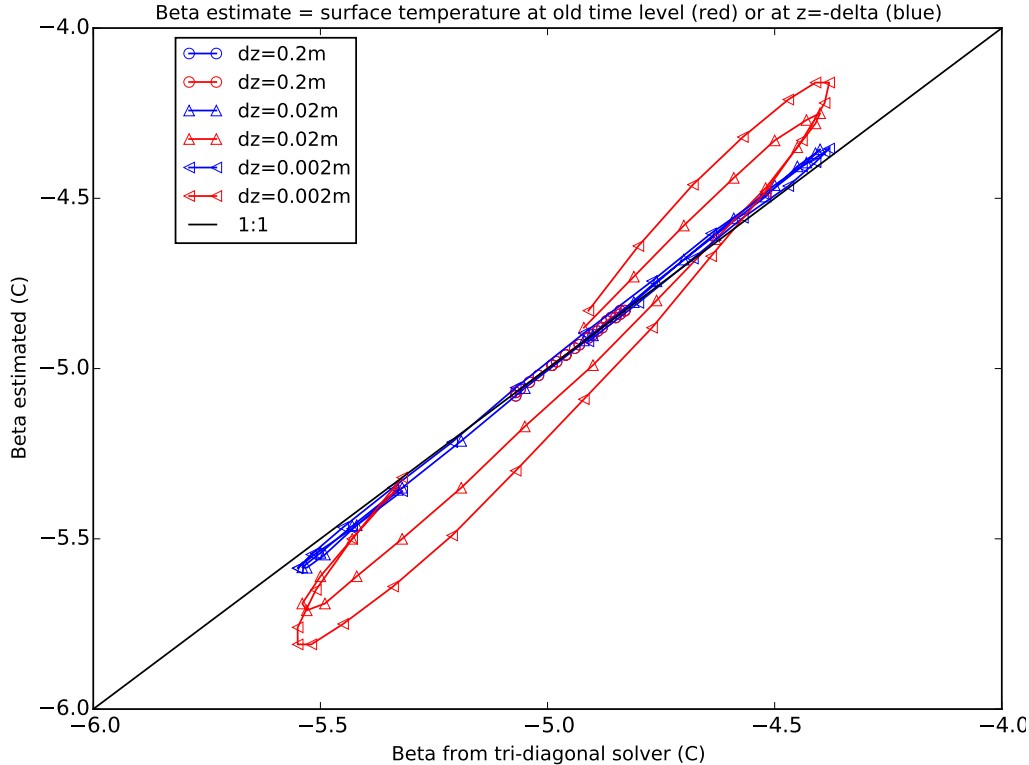

**Figure 7.** Dimensionless $\beta$ as in Fig. 4, but for non-uniform snow density. The snow density is 150 $kgm^{-3}$ at the surface, increases linearly to 250 $kgm^{-3}$ at a depth of 0.5 $m$, and remains constant below 0.5 $m$.

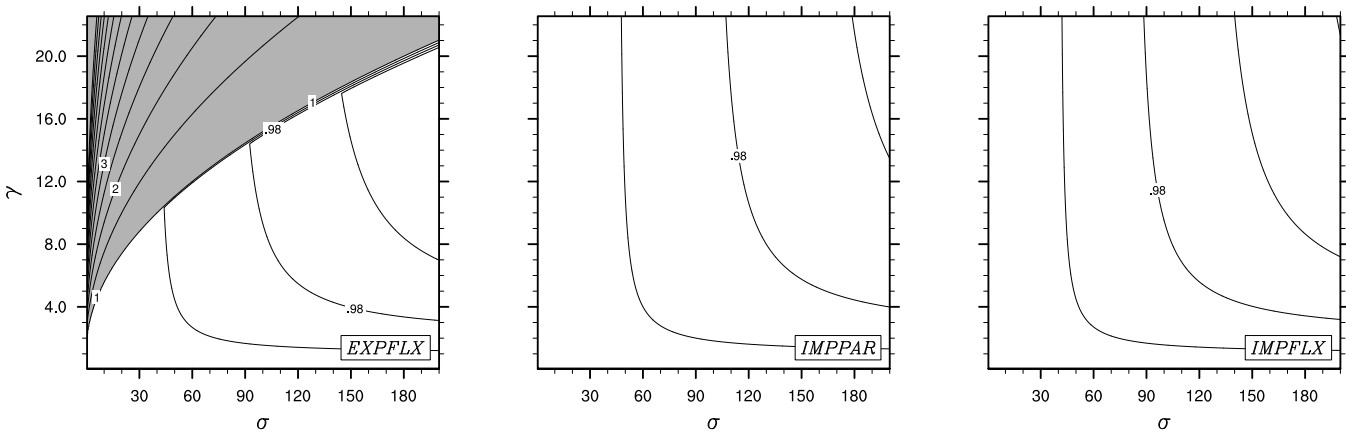

**Figure 8.** Spectral radius of the matrix $\mathbf{M} = \mathbf{A}^{-1}\mathbf{B}$ (defined in B7,B8) associated to the explicit flux coupling (EXPFLX), parametrized implicit flux coupling (IMPPAR), and implicit flux coupling (IMPFLX) with respect to the dimensionless coefficients $\gamma$ and $\sigma$. Gray shaded areas correspond to regions where the spectral radius is larger than 1.

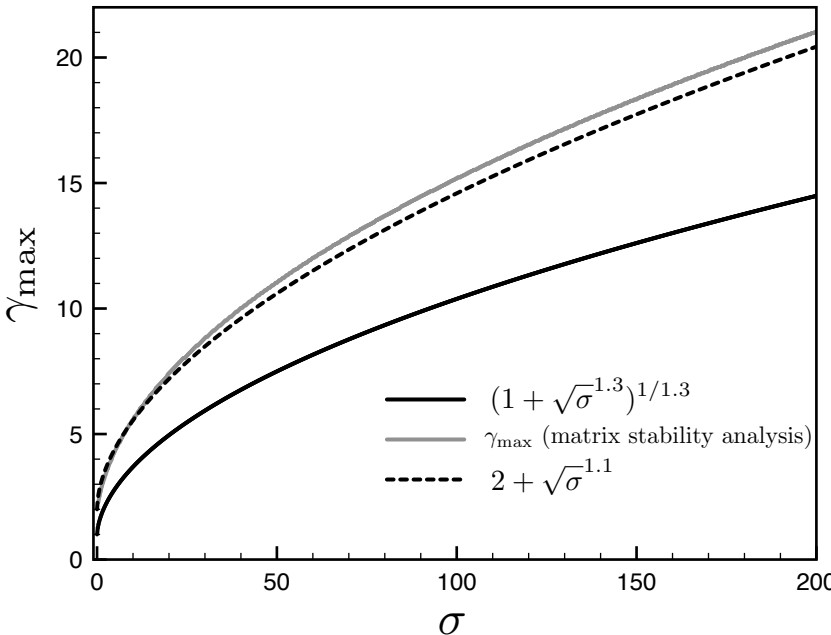

**Figure 9.** Maximum value of $\gamma$ with respect to the parabolic Courant number $\sigma$ to guarantee stability of the explicit flux coupling (solid gray) which roughly behaves like $\gamma(\sigma) = 2 + \sqrt{\sigma}^{1.1}$ (doted black line). The parametrized implicit flux coupling replaces $\gamma$ by $\tilde{\gamma}$ which is always smaller than $\tilde{\gamma}_{max} = (1 + \sqrt{\sigma}^{1.3})^{1/1.3}$ (solid black line). Since the solid black line is below the solid gray line, the implicit flux parametrization is unconditionally stable.