# Peer review of "On the numerical stability of surface-atmosphere coupling in weather and climate models"

_Geoscientific Model Development, 2016_

## Referee Comment (RC1) · F. LEMARIE (Referee) · 14 Jun 2016

**A Review of**
**"*On the numerical stability of surface-atmosphere coupling in weather and climate models*"**
**by Anton Beljaars, Emanuel Dutra and Gianpaolo Balsamo**
* * *
**1   Introduction**

This paper aims at addressing a stability problem found in the coupling between the atmospheric surface layer (ASL) and a snow/ice model. An instability associated to a synchrony problem between the snow/ice model and the surface flux computation is identified and illustrated by simple numerical experiments. A somehow empirical way to suppress this type of instability is introduced and numerically tested. The paper is structured as follows. In Sec. 2, the continuous problem is introduced, it consists of a vertical diffusion equation representing temperature evolution in the snow complemented with a flux (Neumann) surface boundary condition. At a discrete level, this diffusion equation is solved using an Euler backward scheme as usually done in state-of-the-art numerical models. Sec. 3 briefly describes the way to compute the surface flux for the snow model as a function of the atmospheric temperature. This computation is based on the standard semi-empirical Monin-Obukhov (MO) theory. Then, two ways of numerically evaluating this surface flux within an Euler backward framework are presented and referred to as *Explicit flux coupling* (which is the preferred way to proceed in numerical codes) and *Implicit flux coupling*. In Sec. 4, an idealized one-dimensional numerical experiment is used to illustrate the occurrence of a numerical instability when an explicit flux coupling is used jointly with a very fine vertical resolution at the top of the snow model. This instability is absent if an implicit flux coupling is used. In order to control this instability, an extrapolation in time based on physical parameters is presented in Sec. 5 to make a prediction of the surface flux at time level $n + 1$ knowing the state variables of the snow model at time $n$. This ad-hoc extrapolation is built on scaling arguments to estimate the propagation of a surface perturbation over one time-step.

**2   Overall quality of the paper**

This paper raises a very interesting problem which occurs when a diffusion term is constrained by boundary conditions computed using a bulk formulation arising from the MO theory. In this case, instead of the classical Neumann or Dirichlet boundary condition we have to treat a linear combination of both (sometimes called Robin condition). Indeed, boundary condition (3) in the paper can be written as

$$K\partial_z T + \lambda_t T = g(t), \qquad \text{for } z = 0,$$

with $g(t) = \lambda_t T_a$. The paper clearly emphasizes the dilemma we get when discretizing this boundary condition since we expect $K\partial_z T(z = 0)$ and $T(z = 0)$ to be provided at the same moment in time

which is at time level $n+1$ if a backward Euler is used. If this is not the case, a numerical instability can occur even if an unconditionally stable implicit scheme is used to advance the diffusion term. This type of instability is generally unnoticed in the literature because it occurs under very unusual situations. Just for raising this issue and trying to circumvent it, this paper should be considered for publication. The paper is well written and the simple numerical experiments are nicely chosen to illustrate the punchline of the paper. **However I recommend major revisions to make the paper less misleading and more convincing because this issue is important for the modeling community**. The following points must be addressed, because as is the paper has a lack of arguments/proofs of numerical nature. To strengthen the message, I personally think that those proofs should be given in this paper and not in a separate paper with possibly different authors.

**3   General comments**

Some of the comments made in this section are supported by additional material presented in Appendix. This goes beyond the normal reviewer duty but the aim is to illustrate how simple the stability analysis could be to convincingly support the various ideas introduced in the paper.

- The manuscript considers an instability of numerical nature, in this regard we expect a stability analysis to characterize under what circonstances the instability can occur. Following Lemarié et al. (2015) or more simply App. A below, this type of stability analysis is technically affordable and does not require too much theoretical efforts. An important dimensionless number is

$$\gamma = \frac{\lambda_t \Delta t}{\Delta z (\rho C)}$$

which must stay small enough to ensure stability of the coupling between the ASL and the various type of surfaces. Of course all the hypothesis of the Von Neumann stability analysis are not met in this particular problem but this type of analysis always provide useful hints. It could also be interesting to provide in the paper some typical values of the parameter $\gamma$ depending on the surface model (ocean, snow, vegetation, etc) so that the reader can assess how specific to the ASL/snow coupling this problem is. Is it standard to use a vertical resolution $\Delta z$ of the order of $10^{-3}$ m in snow models ?

- The paper could leave the impression that the temporal variation of the atmospheric temperature $T_a$ plays a role in the development of the instability. However it must be clear that the instability occurs even if the atmospheric temperature is held constant in time or is simply set to zero. Hence, this instability can occur in coupled models but also in uncoupled models forced with a bulk formulation.

- The statement in the abstract "*These (instabilities) are due to the choice of large integration time-step, aiming at reducing computational burden*" must be mitigated because it is not the only contributing factor, the vertical resolution or the transfer coefficients value are other important parameters.

- p. 6 line 5, it is adventurous to draw any conclusion on the accuracy of the proposed method based solely on the simple numerical experiments presented in the paper without any well established metric like an $\mathcal{L}^2$ or an $\mathcal{L}^\infty$ norm and a convergence study with spatial/temporal resolution. In this paper, the emphasis is on stability and the current experiments or mathematical analysis do not allow any conclusion on accuracy.

- It is not rigorous enough to assess the efficiency of the proposed empirical coupling method based only on an idealized numerical experiment under very specific conditions. For example, in App. B it is shown that, indeed, the proposed method allows for an unconditionally stable coupling between the ASL and the surface model whatever the parameter values.

- Since this paper is considered for publication in GMD, it would be worthwhile to provide additional

details about the implementation of the proposed method in a numerical model with non-uniform grid and flow-dependent diffusion coefficients.

- In the conclusion, it could be interesting to give some comments on the expected benefits of your approach in realistic models. Besides stability, do you expect significant differences in the physical solutions ?

**4  Technical corrections**

- The way to specify units is inconsistant throughout the paper. For example, units are missing for $\Delta z$ and $\Delta t$ in p. 5 (lines 9 to 20), sometimes units are in italic sometimes not (e.g. p. 9, lines 15-16) ...
- In eqn (19) it should be $T_0$ and not $T_0^n$
- p. 7 line 9, it should be $\Delta z \ll \delta$ and not $\Delta z \ll \delta z$
- In Figure 1, $T_{sk}$ and $T_a$ could be added (instead of $T_{10}$ which is never used in the paper). $\lambda_{sk}$ and $\lambda_a$ could also be reported on the figure.
- In figure 2 the left panels show the skin temperature $T_{sk}$ whereas the left panels of Figure 5 show $T_1$. To facilitate the comparison, the same quantity should be plotted.
- Appendix A is relatively trivial and does not provide useful informations. It could be interesting to use this appendix to be more specific about the elimination and back-substitution steps when solving the tridiagonal problem. We guess a Thomas algorithm is used but it is not explicitly stated.

**A  Stability analysis of the explicit flux coupling**

In the following we consider that $T_a = 0$. It must be clear that this choice does not affect the stability analysis. For personal convenience, we consider here that the vertical grid goes from $k = 1$ at the bottom of the snow model to $k = N$ at the air-snow interface ($T_N$ is thus equivalent to $T_1$ in the paper). Adapting eqn (6) in the paper we obtain

$$T_N^{n+1} = T_N^n - \frac{\Delta t}{\Delta z (\rho C)} \left[ \left( \lambda_t T_N^n + \frac{K}{\Delta z} T_N^{n+1} \right) - \left( \frac{K}{\Delta z} T_{N-1}^{n+1} \right) \right]$$

for a constant diffusion $K$ and grid spacing $\Delta z$. Introducing the dimensionless coefficients

$$\gamma = \frac{\lambda_t \Delta t}{\Delta z (\rho C)}, \qquad \sigma = \frac{K \Delta t}{\Delta z^2 (\rho C)}$$

where $\sigma$ is the standard parabolic (diffusion) Courant number, we end up with the equivalent form

$$T_N^{n+1} = T_N^n - \gamma T_N^n - \sigma T_N^{n+1} + \sigma T_{N-1}^{n+1}. \tag{A.1}$$

Assuming that $T_k^{n+1} = \widehat{T}^{n+1} e^{-ik\theta}$ with $\theta = k_z \Delta z \in [0, \pi]$ a normalized wavenumber in the vertical direction, the $\widehat{T}$'s satisfy

$$\widehat{T}^{n+1} = \widehat{T}^n - \gamma \widehat{T}^n - \sigma \widehat{T}^{n+1} + \sigma \widehat{T}^{n+1} e^{-i\theta},$$

which subsequently provides the following amplification factor $\mathcal{A}$

$$\mathcal{A} = \frac{1 - \gamma}{1 + \sigma (1 - e^{-i\theta})}.$$

The modulus of the amplification factor is less than one as long as

$$\gamma \le 1 + \sqrt{1 + 2\sigma(1+\sigma)(1-\cos\theta)} \qquad \Rightarrow \gamma \le 2, \text{ for } \theta \in [0;\pi]$$

which corresponds to the stability condition of the explicit flux coupling. Note that in general this stability limit corresponds to a conservative condition and we can expect the model to be stable with larger values of $\gamma$ because of the regularizing effect of diffusion below the surface and of the atmospheric response.

**B   Stability analysis of the empirical flux coupling**

Using Eqns (14) and (22) in the paper, we easily find that in the empirical coupling the term $\gamma T_N^n$ in (A.1) is replaced by $\frac{\gamma T_N^n}{1+\alpha\lambda_t}$ with $\alpha = f(\sqrt{\sigma})\sqrt{\frac{\Delta t}{K(\rho C)}}$. Interestingly enough, if $f$ is such that $f(\sqrt{\sigma}) = \sqrt{\sigma}$ we obtain with this modified formulation that $\frac{\gamma T_N^n}{1+\alpha\lambda_t} = \frac{\gamma T_N^n}{1+\gamma}$. This means that in the case $\Delta z \gg \delta$ (i.e. $f(x) = x$) the amplification factor is

$$\mathcal{A} = \frac{1 - \gamma/(1+\gamma)}{1 + \sigma(1 - e^{-i\theta})}$$

whose modulus is always smaller than 1 thus indicating unconditional stability.

**References**

Lemarié, F., Blayo, E., Debreu, L., 2015. Analysis of ocean-atmosphere coupling algorithms: Consistency and stability. Procedia Computer Science 51(0), 2066 – 2075.

---

## Referee Comment (RC2) · A. E. West (Referee) · 29 Jul 2016

**Review of A. Beljaars et al 'On the numerical stability of surface-atmosphere coupling in weather and climate models'**

Alex West, Met Office Hadley Centre

This concise study addresses the problem of instability due to explicit coupling across a surface boundary, and proposes a method by which this instability can be greatly reduced or eliminated. In this method, the coupling is made effectively semi-implicit by allowing the surface / boundary layer solver to make an educated guess of the future bottom boundary condition.

I found this study reasonably well argued and set out, with clear additional evidence provided from one-dimensional simulations, and am persuaded that the method described provides a good approximation to a semi-implicit scheme, with significant improvements over a traditional explicit scheme. The study makes a useful addition to the literature. My only comments relate to ways in which the method might perhaps be explained more clearly and consistently. In particular, I am not sure that the algorithm can be described as acting like a 'fully' implicit coupling scheme (see point 2).

1. **Page 3, line 7: 'The matrix equations can be solved by successive elimination of the C-coefficients from the bottom upward'.**

   This is a crucial step as it provides the initial linear relation between surface flux and top layer temperature; however, I had to work through it quite carefully to understand how this produced equation (8). It is also a little confusingly written as strictly speaking it is the variables that are eliminated, not the coefficients.

   I wonder if it would be worth expanding this line to demonstrate the elimination of bottom layer temperature from the lowest pair of equations, and its result:

   $$A_{NL}T_{NL-1}^{n+1} + B_{NL}T_{NL}^{n+1} = D_{NL} \tag{1}$$

   $$A_{NL-1}T_{NL-2}^{n+1} + B_{NL-1}T_{NL-1}^{n+1} + C_{NL-1}T_{NL}^{n+1} = D_{NL-1} \tag{2}$$

   $$\Rightarrow \quad \frac{A_{NL}}{B_{NL}}T_{NL-1}^{n+1} - \frac{A_{NL-1}}{C_{NL-1}}T_{NL-2}^{n+1} - \frac{B_{NL-1}}{C_{NL-1}}T_{NL-1}^{n+1} = \frac{D_{NL}}{B_{NL}} - \frac{D_{NL-1}}{C_{NL-1}} \tag{3}$$

   Which is an equation of the form

   $$T_{NL-2}^{n+1} = \lambda T_{NL-1}^{n+1} + \mu \tag{4}$$

   It is then easy to see that by repeating upwards, the linear relation $T_1^{n+1} = \alpha G_0 + \beta$ can be obtained, as $G_0$ effectively takes the part of the upper boundary condition (i.e. instead of '$T_0^{n+1}$') in the topmost equation.

2. **Page 4, line 18: 'Together with Eq. (13), $T_1^{n+1}$ and $G_0$ can be computed:' (and following equations, (14), (15).**

It is probably a trivial point, but the solution of equations (14) and (15) actually depends on the future air temperature, $T_a^{n+1}$, already being known. I would guess that it is assumed in the method that the atmospheric simulation is solved first, using bottom boundary conditions from the previous timestep, hence $T_a^{n+1}$ is known when the time comes to calculate the surface flux.

With this understanding, the algorithm works as follows:

i)      Atmospheric simulation is advanced one timestep, using boundary conditions from the previous timestep;
ii)     Implicit approximation to future surface flux is calculated, using new atmospheric temperature and coupling fields received from the snow below according to parameterisation described in Section 5;
iii)    'Future' surface flux is passed to the snow model below;
iv)     Snow model is advanced one timestep, using the future surface flux as its upper boundary condition.

With this algorithm, while the snow model is advanced in time using forcing valid at the **end** of the timestep, the atmosphere model is still advanced in time using forcing valid at the **beginning** of the timestep.  It may just be semantics, but I would argue that this method should be described as semi-implicit, rather than fully-implicit.

The implications of this point for stability of the simulation are probably rather limited. Because of the different timescales under which the atmosphere and snow simulations work, the forcing of the snow on the atmosphere (top layer temperature) would probably be liable to change less rapidly than the forcing of the atmosphere on the snow (surface flux).  An implicitly calculated bottom boundary condition for the atmosphere is probably somewhat less important for stability than the top boundary condition for the snow.

This objection therefore does not represent a fundamental flaw in the method, more really an issue with how it is described.  It would probably be good if the authors could state explicitly somewhere that $T_a^{n+1}$ is solved as part of the atmosphere simulation based on previous values of the top snow layer temperature.

3. **Page 6, line 30 – page 7, line 8 (discussion of scaling behaviour of surface temperature evolation).**

This section (Section 5) contains the central idea of the study, which is impressive in its simplicity.  The idea is based on the fact that dimensional constraints on solutions to the diffusion equation mean that in the case of an initially isothermal profile, the evolution of the surface temperature in response to any surface flux $G_0$ must take a very specific form, described in equation (18) of Beljaars et al.

My issue is that having derived equation (18), the authors appear to digress into a discussion of the surface temperature simulation before arriving at equation (22), which describes the form the coefficient $\alpha$ must take. I think that this is unnecessary (see below). It also appears to conflate two different concepts: the surface skin temperature (SSKT) and top layer temperature (TLT). It is the TLT that the authors are trying to approximate using the scaling arguments, but in equations (19) and (20) they effectively derive the form $\alpha$ would take if they were instead trying to approximate the SSKT. The authors then state that this form must also take account of $\Delta z$ for an approximation of TLT, due to the system being discretised. But I think that the form must take account of $\Delta z$ for the (simpler) reason that it is TLT they are trying to approximate, not SSKT. It is possible that the authors view TLT as being the 'discrete' approximation to SSKT, but I think that it is much more accurate to view it as the discrete approximation to temperature at a depth of $\frac{\Delta z}{2}$.

The same conclusions can be arrived at from equation (18) more smoothly, as follows:

Rearranging equation (18) of Beljaars et al, and rewriting it in the notation of section 2, leads to the form

$$T_j^{n+1} = \frac{\delta G_0}{K} h\left(\frac{z_j}{\delta}\right) + T_j^n \tag{5}$$

Here δ is the scaling depth $\left(\frac{K\Delta t}{\rho C}\right)^{1/2}$, $z_j$ the midpoint of vertical layer $j$, and h() some universal function. It should be noted that (5) still describes the evolution of the temperature according to the continuous diffusion equation, rather than being a description of the discretised solution (even though it is written in this notation).

In particular, setting $j$=1, to estimate the TLT:

$$T_1^{n+1} = \frac{\delta G_0}{K} h\left(\frac{\Delta z}{2\delta}\right) + T_1^n \qquad \text{(using } z_1 = \frac{\Delta z}{2}) \tag{6}$$

Equation (6) shows that in estimating the discretised solution the quantity $\frac{\delta}{K} h\left(\frac{z_j}{\delta}\right)$ naturally takes the place of the coefficient α (the '2' in the denominator of the function argument can be ignored, by scaling $h$ and relabeling). As $h$ is dimensionless, this implies that α scales like $\left(\frac{\Delta t}{K\rho C}\right)^{1/2}$ (the same conclusion as the authors reach in their equation (20)). This shown, the function can be 'measured'.

---

## Author Comment (AC1) · 19 Aug 2016

[gmd, manuscript]copernicus We thank the reviewer (Florian Remarie) for the extensive review, the constructive comments, and for providing a stability analysis to complement the paper. The practical relevance of this paper is clearly recognized and we appreciate the encouragement to publish after revision.

**Response to the general comments**

**1** The paper is about stability of the numerical coupling and we recognize that a formal stability analysis is missing. The reason for not including such an analysis is that the proposed scheme is equivalent to a fully implicit scheme with known stability characteristics. However, we feel that the stability analysis is a welcome addition to the paper.

Using 1 mm thick snow layers is unusual in large scale models. Many models still use a single slab of snow where the layer thickness is controlled by snow mass and density. There are two reasons why the stability issue was urgent in the ECMWF model: (i) in snow accumulation and snow melt conditions the layer thickness can be very small, and (ii) there is evidence that thin layers are needed because the temperature response of the snow skin temperature is rather fast and currently not captured by models. Stability is an absolute requirement for all model points. So far, accuracy implications have been seen from the vertical discretization, but not from the time stepping. Some discussion will be added to the manuscript.

**2** It is indeed interesting that the instability can occur for constant forcing, and the paper might give the wrong impression. We will revise the paper to make sure that this point is clear.

**3** We agree that the statement about long time steps is not very precise. It is of course about the time step in relation to the physical time scale of the discretized problem (which depends on vertical discretization and diffusion coefficients). We will revise the manuscript accordingly.

**4** The statement on p.6 line 5 about accuracy is indeed not very general and limited to the case that is presented. We will modify the text to clarify. It is worth pointing out that the paper is not about accuracy. Numerical accuracy is secondary to

stability for practical applications. Furthermore, we feel that the accuracy of all formulations is good compared to the knowledge about the parametrized equations.

**5** We agree that conclusions on stability can not be generalized on the basis of limited empirical numerical experimentation. However, as pointed out under 1, we were not surprised because the scheme is equivalent to a fully implicit formulation. It is good to see this confirmed by a formal stability analysis.

**6** We agree that it is of interest to consider a non-uniform grid and flow dependent diffusion coefficients. The conclusion already discusses the case of non-uniform diffusion coefficients, and there is no reason why a non-uniform grid would behave differently from non-uniform diffusion coefficients. This point will be added in the discussion.

Temperature dependent diffusion coefficients is a completely different story because it potentially introduces a non-linear instability which is classic in atmospheric boundary layer schemes (see e.g. Kalnay and Kanamitsu 1988: "Time schemes for strongly nonlinear damping equations", Monthly Weather Review, 116, 1945-1958). Discussion of this issue is beyond the scope of the paper.

**7** The benefit for real models is stability (which is an absolute requirement) and that advantage can be taken from finer vertical discretization resulting in a faster response of the surface temperature.

**Response to the technical corrections**

We thank the reviewer for his careful reading. The corrections will be included in the revised manuscript. In line with the request by reviewer 2 we will add a paragraph on

the tri-diagonal solver in Appendix A. It is a standard Gaussian elimination procedure, which I think, is also called the Thomas algorithm.

---

## Author Comment (AC2) · 19 Aug 2016

[gmd, manuscript]copernicus

We thank the reviewer (Alex West) for the comprehensive and constructive review. We very much appreciate the time and effort to understand and digest the proposed method to improve stability. The reviewer finds the method convincing, but feels that the clarity of the presentation can be improved. The comments provided are very helpful.

**Point by point**

**1** Although the solution of the tri-diagonal matrix by Gaussian elimination is standard, it is worth adding a few lines in the Appendix. This suggestion was also made by reviewer 1.

**2** The availability of the forcing temperature at the new time level is a simplification for this study. However, it is not a limitation. The coupling scheme to an atmospheric model (i.e. not just a forcing level) as described by the reviewer, is not what we have in mind. Although it is not the topic of the paper, it is obviously a shortcoming of the paper not to discuss it. Thanks for raising this issue.

The way it can be done in a fully implicit way is by doing the elimination phase of the tri-diagonal matrix for the turbulent diffusion in the atmosphere (from top to surface) exactly in same way as is done for the surface heat diffusion in this paper. This procedure leads also to a linear relation between temperature at the lowest atmospheric model level and the heat flux into the surface. This relation replaces the imposed temperature at the new time level. This is precisely the coupling procedure followed in the ECMWF model and is compatible with the Best et al. (2004) approach. It is equivalent to solving a single tri-diagonal matrix that handles the entire atmosphere and surface as a single implicit problem.

We propose to add a paragraph in the paper to discuss how the method can be applied in a fully coupled atmosphere / surface model.

**3** The reviewer raises an interesting point on the temporal evolution of temperature as the result of a perturbation at the surface. It is argued that the final scaling relation can be achieved in a more logical way. Reading the manuscript again, we agree that the transition from the continuous to the discrete problem is vague, which does not help clarity of the manuscript.

The simplest way of deriving a similarity relation for $\alpha$ is by doing a basic dimension analysis, which leads directly to equation (20) of the manuscript for the continuous problem (given time scale $\Delta t$). The discrete problem adds a new length scale resulting in a functional dependence on $(\delta/\Delta z)$, i.e. equation (22) of the manuscript. We opted to start with the traditional scaling relation for the diffusion equation and then to add the complexity of the discretization.

The derivation by the reviewer is attractive, but after thinking about it more carefully, I am not sure any more. The attractive aspect is the simplicity and the fact that function h (eq. 20 of the manuscripts) comes back as function f (eq. 22). It is important to realize that the meaning is not necessarily the same. Function h represents the shape of the temperature profile, whereas function f is an empirical function that describes the transition from one asymptotic scaling regime (the non-discretized problem) to another (the extremely coarse vertical resolution regime). The two functions are related and perhaps even close, but I am not sure they are the same. To demonstrate the potential difference, it is necessary to consider the average temperature (or average h) over the discretization intervals, instead of using midpoint values, otherwise conservation is lost (I think).

The beauty of similarity theory is that we don't need to answer this question. Function f is an empirical function and we can derive it from numerical experiments.

In view of the discussion above, we propose to modify the manuscript such that it is clear where the transition is made from the continuous to the discrete problem.

---

## Author Response (AR1)

The authors thank both referees for the extensive reviews, constructive comments, and for providing many suggestions for improvement. The practical relevance of this paper is clearly recognized and we appreciate the encouragement to publish after revision.

**Revisions suggested by reviewer 1**

**Introduction**

*... a numerical instability can occur even if an unconditionally stable implicit scheme is used to advance the diffusion term. This type of instability is generally unnoticed in the literature because it occurs under very unusual situations. Just for raising this issue and trying to circumvent it, this paper should be considered for publication. The paper is well written and the simple numerical experiments are nicely chosen to illustrate the punchline of the paper. However I recommend major revisions to make the paper less misleading and more convincing because this issue is important for the modeling community. The following points must be addressed, because as is the paper has a lack of arguments/proofs of numerical nature. To strengthen the message, I personally think that those proofs should be given in this paper and not in a separate paper with possibly different authors.*

The manuscript has been revised in a major way by addressing the misleading aspects, and by adding a stability analysis in Appendix B.

**General comments, bullet 1**

*The manuscript considers an instability of numerical nature, in this regard we expect a stability analysis to characterize under what circonstances the instability can occur.*

*It could also be interesting to provide in the paper some typical values of the parameter $\gamma$.*

*Is it standard to use a vertical resolution $\Delta z$ of the order of 0.001 m in snow models ?*

Reviewer 1 recognizes the practical relevance of the paper, but feels that a formal stability analysis is missing. We agree that a stability analysis is a welcome addition to the paper. For this purpose a "matrix stability analysis" has been added as Appendix B. It confirms that the implicit coupling and the parametrized implicit coupling are unconditionally stable.

A new table 2 with values for $\gamma$ and $\sigma$ has been included.

Snow models with 0.001 m vertical resolution are rare, but models must be able to cope with such thin layers, e.g. during the early accumulation of snow and the final melt. Reference is made now to such situations in the introduction.

**General comments, bullet 2**

*The paper could leave the impression that the temporal variation of the atmospheric temperature Ta plays a role in the development of the instability. However it must be clear that the instability occurs even if the atmospheric temperature is held constant in time or is simply set to zero. Hence, this instability can occur in coupled models but also in uncoupled models forced with a bulk formulation.*

With the stability analysis of Appendix B, it is clear now that numerical stability does not depend on the forcing, because it assumes zero perturbations at the forcing level.

**General comments, bullet 3**

*The statement in the abstract "These (instabilities) are due to the choice of large integration time-step, aiming at reducing computational burden" must be mitigated because it is not the only contributing factor, the vertical resolution or the transfer coefficients value are other important parameters.*

We agree that the statement about long time steps is not very precise. It is of course about the time step in relation to the physical time scale of the discretized problem (which depends on vertical discretization and properties of the medium). The abstract has been revised accordingly.

**General comments, bullet 4**

*p. 6 line 5, it is adventurous to draw any conclusion on the accuracy of the proposed method based solely on the simple numerical experiments presented in the paper ....*

The statement on p.6 line 5 about accuracy is indeed not very general and limited to the case that is presented. A comment of this nature has been added at the beginning of the previous paragraph.

**General comments, bullet 5**

*It is not rigorous enough to assess the efficiency of the proposed empirical coupling method based only on an idealized numerical experiment under very specific conditions.*

We agree that conclusions on stability can not be generalized on the basis of limited numerical experimentation. A stability analysis has been added in Appendix B, to which reference is made at various places in the manuscript.

**General comments, bullet 6**

*Since this paper is considered for publication in GMD, it would be worthwhile to provide additional details about the implementation of the proposed method in a numerical model with non-uniform grid and flow-dependent diffusion coefficients.*

We agree that it is of interest to consider a non-uniform grid and flow dependent diffusion coefficients. The concluding section already discusses the case of non-uniform diffusion coefficients, and in fact the aerodynamic coupling between atmosphere and snow can just be seen as an extreme jump in properties of the medium. This point has been added in the discussion.

Temperature dependent diffusion coefficients is a completely different story because it potentially introduces a non-linear instability which is classic in atmospheric boundary layer schemes (see e.g. Kalnay and Kanamitsu 1988: "Time schemes for strongly nonlinear damping equations", Monthly Weather Review, 116, 1945-1958). Discussion of this issue is beyond the scope of the paper.

**General comments, bullet 7**

*In the conclusion, it could be interesting to give some comments on the expected benefits of your approach in realistic models. Besides stability, do you expect significant differences in the physical solutions ?*

The benefit for real models is stability (which is an absolute requirement) and advantage can be taken from finer vertical discretization resulting in a faster response of the surface temperature. The first paragraph of the concluding section has been rewritten.

**Technical corrections, bullet 1**

*The way to specify units is inconsistent throughout the paper.*

Units are italic now throughout.

**Technical corrections, bullet 2**

*In eqn (19) it should be $T_0$ and not $T_0^n$*

Equation (19) has been deleted to improve clarity in response to reviewer 2.

**Technical corrections, bullet 3**

*p. 7 line 9, it should be $\Delta z << \delta$ and not $\Delta z << \delta z$*

Has been corrected.

**Technical corrections, bullet 4**

*In Figure 1, $T_{sk}$ and $T_a$ could be added (instead of $T_{10}$ which is never used in the paper). $\lambda_{sk}$ and $\lambda_a$ could also be reported on the figure.*

Figure 1 has been adapted.

**Technical corrections, bullet 5**

*In figure 2 the left panels show the skin temperature $T_{sk}$ whereas the left panels of Figure 5 show $T_1$. To facilitate the comparison, the same quantity should be plotted.*

Fig. 5 shows $T_{sk}$ now.

**Technical corrections, bullet 6**

*Appendix A is relatively trivial and does not provide useful informations. It could be interesting to use this appendix to be more specific about the elimination and back-substitution steps when solving the tridiagonal problem. We guess a Thomas algorithm is used but it is not explicitly stated.*

A paragraph has been added in appendix A to explain the tri-diagonal solver. It is a standard Gaussian elimination procedure, which I think, is also called the Thomas algorithm.

**Revisions suggested by reviewer 2**

**Introduction**

*I found this study reasonably well argued and set out, with clear additional evidence provided from one-dimensional simulations, and am persuaded that the method described provides a good approximation to a semi-implicit scheme, with significant improvements over a traditional explicit scheme. The study makes a useful addition to the literature. My only comments relate to ways in which the method might perhaps be explained more clearly and consistently. In particular, I am not sure that the algorithm can be described as acting like a "fully" implicit coupling scheme (see point 2).*

The suggestion to improve clarity were very helpful. The different points are described below.

1. **Page 3, line 7: The matrix equations can be solved by successive elimination of the C-coefficients from the bottom upward.**

   *This is a crucial step as it provides the initial linear relation between surface flux and top layer temperature; however, I had to work through it quite carefully to understand how this produced equation (8). It is also a little confusingly written as strictly speaking it is the variables that are eliminated, not the coefficients. I wonder if it would be worth expanding this line to demonstrate the elimination of bottom layer temperature from the lowest pair of equations, and its result*

   The solution of the tri-diagonal matrix by Gaussian elimination is fairly standard. The details of the elimination process have been added in Appendix A.

2. **Page 4, line 18: "Together with Eq.(13) $T_1^{n+1}$ and $G_0$ can be computed:" (and following equations (14,(15)).**

   *It is probably a trivial point, but the solution of equations (14) and (15) actually depends on the future air temperature, $T_a^{n+1}$, already being known. I would guess that it is assumed in the method that the atmospheric simulation is solved first, using bottom boundary conditions from the previous time step, hence $T_a^{n+1}$ is known when the time comes to calculate the surface flux.*

   The availability of the forcing temperature at the new time level is a simplification for this study. However, it is not a limitation. The coupling scheme to an atmospheric model (i.e. not just a forcing level) as described by the reviewer, is not what we have in mind. Although it is not the topic of the paper, it is obviously a shortcoming of the paper not to discuss it. Thanks for raising this issue.

The way it can be done in a fully implicit way is by doing the elimination phase of the tri-diagonal matrix for turbulent diffusion in the atmosphere (from top to surface) exactly in same way as is done for the surface heat diffusion in this paper. This procedure leads also to a linear relation between temperature at the lowest atmospheric model level and the heat flux into the surface. This relation replaces the imposed temperature at the new time level. This is precisely the coupling procedure followed in the ECMWF model and is compatible with the Best et al. (2004) approach. It is equivalent to solving a single tri-diagonal matrix that handles the entire atmosphere and surface as a single implicit problem.

A paragraph has been added in the paper to discuss how the method can be applied in a fully coupled atmosphere / surface model.

**3. Page 6, line 30 - page 7, line 8 (discussion of scaling behavior of surface temperature evolution).**

*This section (Section 5) contains the central idea of the study, which is impressive in its simplicity. The idea is based on the fact that dimensional constraints on solutions to the diffusion equation mean that in the case of an initially isothermal profile, the evolution of the surface temperature in response to any surface flux must take a very specific form, described in equation (18) of Beljaars et al. My issue is that having derived equation (18), the authors appear to digress into a discussion of the surface temperature simulation before arriving at equation (22), which describes the form the coefficient must take. I think that this is unnecessary (see below). It also appears to conflate two different concepts: the surface skin temperature (SSKT) and top layer temperature (TLT). It is the TLT that the authors are trying to approximate using the scaling arguments, but in equations (19) and (20) they effectively derive the form would take if they were instead trying to approximate the SSKT. The authors then state that this form must also take account of $\Delta z$ for an approximation of TLT, due to the system being discretized. But I think that the form must take account of $\Delta z$ for the (simpler) reason that it is TLT they are trying to approximate, not SSKT. It is possible that the authors view TLT as being the "discrete" approximation to SSKT, but I think that it is much more accurate to view it as the discrete approximation to temperature at a depth of $\Delta z/2$*

The reviewer raises an interesting point on the temporal evolution of temperature as the result of a perturbation at the surface. It is argued that the final scaling relation can be achieved in a more logical way. Reading the manuscript again, we agree that the transition from the continuous to the discrete problem is vague, which does not help clarity of the manuscript.

The simplest way of deriving a similarity relation for $\alpha$ is by doing a basic dimension analysis, which leads directly to equation (19) of the revised manuscript for the continuous problem (given time scale $\Delta t$). The discrete problem adds a new length scale resulting in a functional dependence on $(\delta/\Delta z)$, i.e. equation (21) of the revised manuscript. We opted to start with the traditional scaling relation for the diffusion equation and then to add the complexity of the discretization.

The derivation by the reviewer is attractive, but after thinking about it more carefully, I am not sure any more. The attractive aspect is the simplicity and the fact that function h (eq. 18 of the manuscripts) comes back as function f (eq. 21). It is important to realize that the meaning is not necessarily the same. Function h represents the shape of the temperature profile, whereas function f is an empirical function that describes the transition from one asymptotic scaling regime (the non-discretized problem) to another (the extremely coarse vertical resolution regime). The two functions are related and perhaps even close, but I am not sure they are the same. To demonstrate the potential difference, it is necessary to consider the average temperature (or average h) over the discretization intervals, instead of using midpoint values, otherwise conservation is lost (I think).

The beauty of similarity theory is that we don't need to answer this question. Function f is an empirical function and we can derive it from numerical experiments.

In view of the discussion above, the manuscript has been modified such that it is clear where the transition is made from the continuous to the discrete problem.

[revised manuscript text omitted]

---

## Referee Report (RR1)

**Second review of Beljaars et al, 'On the numerical stability of surface-atmosphere coupling in weather and climate model'**

Alex West, Met Office Hadley Centre

This revised study describes a method of approximating implicit coupling between surface exchange and the underlying medium.  It differs principally from the first version in that in addition to evidence from 1-dimensional experiments, it contains mathematical proof that the method described is, like truly implicit coupling, unconditionally stable, irrespective of the forcing atmospheric temperatures used.  In addition, some aspects of the presentation have been clarified.

The authors have addressed my previous concerns about clarity well.  My 'point 2' about the atmospheric air temperatures needing to be known for the solution of equations (14) and (15) was based on a misunderstanding (forced versus coupled simulation), and the authors have ensured this will not be repeated with their revised introduction to section 3.  The revised section 5 also reads much better.  I take the authors' point about the temperature at depth $\Delta z/2$ being a slightly different concept to the modelled temperature of the grid cell centred at $\Delta z/2$.

The new matrix stability analysis, illustrated with Figures 8 and 9, is a very useful addition to the paper, and appears to be well reasoned.  Florian Lemarié's point that instability could occur even with constant air temperature was very illuminating.  The authors are correct in stating that temperature-dependent diffusion is a much more complex problem and beyond the scope of the paper.

I have only one revision to suggest: with the altered reasoning of section 5, the variable $h_0$ is used on page 7, line 18 without having been defined.  It might be a good idea if this sentence was either deleted, or reworded: 'Surprisingly, the function asymptotes to 1'.  Alternatively, $h_0$ could be redefined further up.

With this addressed, the manuscript can be published.

---

## Referee Report (RR2)

**Review of "On the numerical stability of surface-atmosphere coupling in weather and climate models" by Beljaars et al.**

It was a pleasure to read this paper and also a surprise to see that this issue has not yet been solved in all land-surface models. This is the typical problem of flux equation discretizations at discontinuities. I have also recently leaned that similar methods are being used for solving diffusion equations on parallel computers when the domain decomposition cuts through the diffusion.

It is not true that an implicit coupling is incompatible with a modular code structure. This has been discussed and solved since Polcher at al. 1998. Most European weather and climate model use a fully implicit coupling and have a modular structure which allows them to run their land-surface scheme off-line or the atmosphere as an aqua-planet without changing the code. Recently ORCHIDEE was coupled to WRF using OASIS while maintaining the implicit solution to the full set of vertical diffusion equations ! With some of the authors we have revisited the issue a few years ago and it is surprising that the fully implicit coupling is still not implemented at ECMWF !

I do not think that introducing an empirical equation is needed to link $T_1$ to the heat flux at the skin temperature level. During our thesis with Frédréric Hourdin (Hourdin 1992), we proposed to extrapolate the two upper soil temperatures towards the interface and thus provide the surface energy balance equation with a flux and a heat capacity for the infinitesimal layer for which the skin temperature is computed. The benefit of this methodology for complex land surface scheme was documented during the thesis of Jan-Peter Schulz (Schulz et al. 2001).

This has recently been extended to the multi-layer snow scheme without problems (Wang et al. 2013). The referenced paper covers the numerically explicit version of the scheme but this was converted to an implicit scheme coupled to the surface energy balance equation without problems (Will be described in the ORCHIDEE documentation for CMIP6). The same extrapolation then also needs to be done at the bottom of the snow in order to link the heat diffusion in the snow pack with the diffusion in the soil while maintaining a temperature at the contact point between the two media. This allows to solve implicitly the diffusion equations from the top of the atmosphere to the bottom of the soil with clear interfaces and a flexible modular code.

We have also verified that the thermal diffusion equaion is numerically stable for very thin upper layers in the soil. This needed to be done when we decided to use a common vertical discretization for the hydrology and thermodynamics (Wang et al 2016). Thus we are quite satisfied with this approach to combine the diffusion equation with an interface temperature. We would agree that the linearity assumption can be criticized on physical grounds.

Indeed, the first few millimetres away from the interface temperature are characterized by very strong gradients and changes in properties which probably break any linearity assumption. Just as a single skin temperature for a complex canopy is dubious. We are currently progressing toward a true 3D energy balance which takes into account the complexity of the vertical structure of the medium which interacts with the atmosphere (Ryder et al. 2015). Even such complex set of equations can be simply solved implicitly and with keeping a maximum of modularity in the code.

I cannot recommend the publication of this paper as it does not take into account sufficiently previous publications on this topic. The references I provide below are only the one I know well as I work everyday with that code. But I know that at Princeton and NCAR some very interesting progress has been made recently on the numerics of the surface processes but I would not know what the most pertinent reference is. Our colleagues at MPI Hamburg have also developed some interesting methods for coupling implicitly multiple surface energy balances to atmospheric columns which are relevant to the discussion proposed.

I do not think either that introducing the empirical equation 8 and its two parameters is justified when the fully implicit solution can be implemented in a modular way and the interface temperature issue solved more elegantly. The linear relation assumed in the Hourdin method is much more tolerable than the scaling relation for $\alpha$ and $\beta$ !

Jan Polcher

**References :**

Hourdin, F.: Etude et simulation numérique de la circulation générale des atmosphères planétaires, PhD Thesis, available at: www.lmd.jussieu.fr/~hourdin/these.pdf (last access: 24 January 2016), 1992.

Polcher, J., McAvaney, B., Viterbo, P., Gaertner, M.-A., Hahmann, A., Mahfouf, J.-F., Noilhan, J., Phillips, T., Pitman, A.J., Schlosser, C.A., Schulz, J.-P., Timbal, B., Verseghy D., and Xue, Y. (1998) A proposal for a general interface between land-surface schemes and general circulation models. *Global and Planetary Change*, **19**:263-278.

J. Ryder, J. Polcher, P. Peylin, C. Ottlé, Y. Chen, E. van Gorsel, V. Haverd, M. J. McGrath, K. Naudts, J. Otto, A. Valade, and S. Luyssaert (2015) A multi-layer land surface energy budget model for implicit coupling with global atmospheric simulations, Geosci. Model Dev., 9, 223-245, doi:10.5194/gmd-9-223-2016.

Schulz JP, Dümenil L, Polcher J (2001) On the land surface-atmosphere coupling and its impact in a single-column atmospheric model. J Appl Meteorol 40(3):642–663
Wang, F., Cheruy, F., and Dufresne, J.-L.: The improvement of soil thermodynamics and its effects on land surface meteorology in the IPSL climate model, Geosci. Model Dev., 9, 363-381, doi:10.5194/gmd-9-363-2016, 2016.

Wang, T., C. Ottlé, A. Boone, P. Ciais, E. Brun, S. Morin, G. Krinner, S. Piao, and S. Peng (2013), Evaluation of an improved intermediate complexity snow scheme in the ORCHIDEE land surface model, J. Geophys. Res. Atmos., 118, 6064–6079, doi:10.1002/jgrd.50395.

---

## Author Response (AR2)

**Reviewer report**

It was a pleasure to read this paper and also a surprise to see that this issue has not yet been solved in all land-surface models. This is the typical problem of flux equation discretizations at discontinuities. I have also recently leaned that similar methods are being used for solving diffusion equations on parallel computers when the domain decomposition cuts through the diffusion.

5     It is not true that an implicit coupling is incompatible with a modular code structure. This has been discussed and solved since Polcher at al. 1998. Most European weather and climate model use a fully implicit coupling and have a modular structure which allows them to run their land-surface scheme off-line or the atmosphere as an aqua-planet without changing the code. Recently ORCHIDEE was coupled to WRF using OASIS while maintaining the implicit solution to the full set of vertical diffusion equations ! With some of the authors we have revisited the issue a few years ago and it is surprising that the fully

10   implicit coupling is still not implemented at ECMWF !
    I do not think that introducing an empirical equation is needed to link T1 to the heat flux at the skin temperature level. During our thesis with Frédréric Hourdin (Hourdin 1992), we proposed to extrapolate the two upper soil temperatures towards the interface and thus provide the surface energy balance equation with a flux and a heat capacity for the infinitesimal layer for which the skin temperature is computed. The benefit of this methodology for complex land surface scheme was documented

15   during the thesis of Jan-Peter Schulz (Schulz et al. 2001).
    This has recently been extended to the multi-layer snow scheme without problems (Wang et al. 2013). The referenced paper covers the numerically explicit version of the scheme but this was converted to an implicit scheme coupled to the surface energy balance equation without problems (Will be described in the ORCHIDEE documentation for CMIP6). The same extrapolation then also needs to be done at the bottom of the snow in order to link the heat diffusion in the snow pack with the diffusion in the

20   soil while maintaining a temperature at the contact point between the two media. This allows to solve implicitly the diffusion equations from the top of the atmosphere to the bottom of the soil with clear interfaces and a flexible modular code.
    We have also verified that the thermal diffusion equation is numerically stable for very thin upper layers in the soil. This needed to be done when we decided to use a common vertical discretization for the hydrology and thermodynamics (Wang et al 2016). Thus we are quite satisfied with this approach to combine the diffusion equation with an interface temperature. We

25   would agree that the linearity assumption can be criticized on physical grounds.
    Indeed, the first few millimetres away from the interface temperature are characterized by very strong gradients and changes in properties which probably break any linearity assumption. Just as a single skin temperature for a complex canopy is dubious. We are currently progressing toward a true 3D energy balance which takes into account the complexity of the vertical structure of the medium which interacts with the atmosphere (Ryder et al. 2015). Even such complex set of equations can be simply

30   solved implicitly and with keeping a maximum of modularity in the code.
    I cannot recommend the publication of this paper as it does not take into account sufficiently previous publications on this topic. The references I provide below are only the one I know well as I work everyday with that code. But I know that at Princeton and NCAR some very interesting progress has been made recently on the numerics of the surface processes but I would not know what the most pertinent reference is. Our colleagues at MPI Hamburg have also developed some interesting

35   methods for coupling implicitly multiple surface energy balances to atmospheric columns which are relevant to the discussion proposed.
    I do not think either that introducing the empirical equation 8 and its two parameters is justified when the fully implicit solution can be implemented in a modular way and the interface temperature issue solved more elegantly. The linear relation assumed in the Hourdin method is much more tolerable than the scaling relation for $\alpha$ and $\beta$ !

40   **Author's response to reviewer report**

*We thank the reviewer (Jan Polcher) for his review and for pointing to the literature on implicit coupling. It was good to hear that the paper has been pleasurable reading. The main concern expressed in the review, is the lack of reference to existing literature on full implicit coupling. We agree that the lack of reference to existing literature is an oversight, and propose a revision of the introduction as indicated in the marked-up version of the manuscript. A few further minor changes are made in*

45   *the text where reference is made to code modularity.*

*I agree mostly with what is said in the review, but feel that the reviewer is overly optimistic about what is practical in operational models. Such systems have to perform well globally, have to be fast, work with long/variable time steps, and code has to be maintained and managed with very few people.*

*The current manuscript does by no means question the benefit of fully implicit coupling. It just offers an elegant solution for a stability problem in models that do not have fully implicit coupling on all vertical levels. I have no complete overview of European models, but I doubt that most models use fully implicit coupling.The ECMWF model is certainly not the only model with a "break in implicitness" below the fast skin level (see e.g. West et al. 2016 about JULES). In fact the West et al. (2016) paper was the inspiration for writing the current paper, because it was felt that a simple alternative was possible.*

*We all know that different modeling groups are making different choices regarding complexity. ECMWF has opted for a tiled surface scheme to represent surface heterogeneity and plug-compatibility with implicit tiled coupling with the fastest responding land surface layer (namely the skin layer) as addressed in the Best et al. (2004) paper. Keeping full implicitness further down, would require to perform the elimination sweeps of the tridiagonal problems for snow, sea ice, snow, soil and lakes before coupling to the skin layer. It also requires a reformulation of the snow melt and soil freezing through the use of a total energy variables (conserved with respect to freezing/melting). This is certainly do-able but requires major code development. The current paper just offers a practical solution for stability in models that need it, and I am sure that there are more (if not most) models that occasionally hit stability issues, particularly when more realism is introduces regarding fast responding layers near the surface (e.g. thin snow layers).*

*The last sentence of the review is probably based on a misunderstanding. Equation (8) is an empirical form but within the accuracy of the empirical representation, it is exact for the problem under study. In other words it is very accurate given the vertical discretization. The "linear approximation" in Hourdin is about the vertical discretization and as far as I can see not about the time stepping. We probably disagree about what is tolerable in the vertical discretization. I obviously prefer the skin layer formulation in the ECMWF model (as in the current manuscript) because it derives the skin temperature from the properties of the medium between the middle of the layer and the top. Although the accuracy is not discussed in a comprehensive way, simulations in the current paper suggest that the formulation is quite accurate even for thick snow layers. The skin temperature also responds well to fast changes in the forcing. Again, this is not the topic of the paper. The paper is about stability of the time stepping with empirical implicit coupling.*

[revised manuscript text omitted]

---

## Author Response (AR3)

**Editor report**

Dear Author,

Thank you for your updated manuscript and reply to referee 2's comment, which I find convincing. I just have the following very minor suggestions:

5  - P1, L6: I would replace "surface models' integrations" by "surface model integrations"

- P1, L17: I would remove the comma in "... surface model, involves ..."

- P2, L20: I would change "dependent" for "depending"

- P2, L21: I would remove the comma in "... in this paper, is a simple ..."

- P9, L17: I would remove the comma in "... used above, only applies ..."

10  - P9, L18: please change "... there may a profile ..." for "... there may be a profile ..."

With best regards, Sophie Valcke, topical editor

**Author's response to editor**

We thank the editor for her positive response and for the suggestions for correction. All the suggested corrections have been made in the revised manuscript.

15  The following additional changes were made:

- The new affiliation of the second author has been added.

- The project support of the 4th author is mentioned now in the Acknowledgements.